# TFPI from erythroblasts drives heme production in central macrophages promoting erythropoiesis in polycythemia

Jun-Kai Ma[1,12], Li-Da Su [2,12], Lin-Lin Feng[1,3,12], Jing-Lin Li[1], Li Pan[4], Qupei Danzeng[5], Yanwei Li[6], Tongyao Shang [1], Xiao-Lin Zhan[1,3], Si-Ying Chen[1,3], Shibo Ying [7], Jian-Rao Hu[8], Xue Qun Chen [9], Qi Zhang [10,11]✉, Tingbo Liang [10,11]✉ & Xin-Jiang Lu [1]✉

Bleeding and thrombosis are known as common complications of polycythemia for a long time. However, the role of coagulation system in erythropoiesis is unclear. Here, we discover that an anticoagulant protein tissue factor pathway inhibitor (TFPI) plays an essential role in erythropoiesis via the control of heme biosynthesis in central macrophages. TFPI levels are elevated in erythroblasts of human erythroblastic islands with $JAK2^{V617F}$ mutation and hypoxia condition. Erythroid lineage-specific knockout TFPI results in impaired erythropoiesis through decreasing ferrochelatase expression and heme biosynthesis in central macrophages. Mechanistically, the TFPI interacts with thrombomodulin to promote the downstream ERK1/2-GATA1 signaling pathway to induce heme biosynthesis in central macrophages. Furthermore, TFPI blockade impairs human erythropoiesis in vitro, and normalizes the erythroid compartment in mice with polycythemia. These results show that erythroblast-derived TFPI plays an important role in the regulation of erythropoiesis and reveal an interplay between erythroblasts and central macrophages.

Adult humans produce 2 to 3 million red blood cells (RBCs) every second in bone marrow (BM) during steady state erythropoiesis, which is challenged by multiple variables such as $JAK2^{V617F}$ mutation and high altitude hypoxia[1,2]. The clinical course of polycythemia is marked by the high incidence of bleeding and thrombosis[3]. Tissue factor (TF) is best known as the primary cellular initiator of blood coagulation. TF pathway inhibitor (TFPI) is the principal inhibitor of the initiation of blood coagulation through high-affinity TF inhibition[4]. TF activity is increased in neutrophils from polycythemia vera patients[5]. These results establish abnormal coagulation as an important physiological process in polycythemia, in which the underlying molecular mechanisms affecting erythropoiesis by coagulation factors are unknown.

The process of erythropoiesis is the largest consumer of iron, which is employed by erythroid lineages to synthesize heme as a

[1]Department of Physiology and Department of Hepatobiliary and Pancreatic Surgery of the First Affiliated Hospital, Zhejiang University School of Medicine, Hangzhou 310058, China. [2]Neuroscience Care Unit, Second Affiliated Hospital of Zhejiang University School of Medicine, Hangzhou 310009, China. [3]State Key Laboratory for Quality and Safety of Agro-products, Ningbo University, Ningbo 315211, China. [4]The General Hospital of Tibet Military Area Command, Lhasa, China. [5]Department of Tibetan Medicine; University of Tibetan Medicine, Lhasa 540100, China. [6]Core Facilities, Zhejiang University School of Medicine, Hangzhou, China. [7]School of Public Health, Hangzhou Medical College, Hangzhou 310013, China. [8]School of Life and Environmental Sciences, Hangzhou Normal University, Hangzhou 311121 Zhejiang, China. [9]Zhejiang University, School of Brain Science and Brain Medicine, Hangzhou, China. [10]Department of Hepatobiliary and Pancreatic Surgery, the First Affiliated Hospital, Zhejiang University School of Medicine, Hangzhou 310058, China. [11]Zhejiang Provincial Key Laboratory of Pancreatic Disease, the First Affiliated Hospital, Zhejiang University School of Medicine, Hangzhou 310058, China. [12]These authors contributed equally: Jun-Kai Ma, Li-Da Su, Lin-Lin Feng. ✉e-mail: qi.zhang@zju.edu.cn; liangtingbo@zju.edu.cn; lxj711043@163.com

structural component of hemoglobin of erythroid precursors[6]. Intercellular heme transportation may play an important role in erythropoiesis. Macrophages express feline leukemia virus subgroup C receptor (FLVCR), a heme export protein[7]. Hematopoietic cell-specific FLVCR knockout mice display severe erythropoietic deficiency[8]. Heme released from degraded RBCs in macrophage is exported as intact heme through heme exporters[7]. Moreover, heme carrier protein 1 is responsible for exogenous heme import, which mediates erythroblasts differentiation to red cells in BM[9], suggesting that exogenous heme may contribute to erythropoiesis. It is possible that non-erythroid lineage-derived heme contributes to erythropoiesis during steady-state or stress conditions.

Erythropoiesis requires a specific microenvironment comprised of central macrophages surrounded by developing erythroblasts[10]. Central macrophages act as nurse cells for erythroblasts and are integral components of erythroblastic islands (EBIs), which were firstly identified in bone marrow[10-13]. Later investigation identified EBIs in the murine spleen and fetal liver, highlighting the essential nature of central macrophages for promoting erythropoiesis[14]. Unlike other macrophages, central macrophages promote erythropoiesis by providing nutrients and signals to surrounding erythroblasts, as well as phagocytosing nuclei extruded from developing erythrocytes[15,16]. The interaction between central macrophages and erythroblasts is mediated by adhesion molecules, such as Vcam-1 and CD169 on macrophages[17]. Macrophages produce bone morphogenetic protein 4 to promote erythropoietic recovery following myeloablation[11]. Furthermore, splenic macrophages degrade heme and recycle the iron for de novo erythropoiesis[18]. Macrophages express mechanosensitive piezo1 to affect ion metabolism and subsequent erythrocyte turnover[19]. Moreover, erythroblasts also secrete regulatory factors to affect hematopoietic progenitor cells or iron loading. Erythroblast-derived fibroblast growth factor 23 is needed to release hematopoietic progenitor cells from BM into the circulation[20]. The erythroid lineage-derived hormone erythroferrone is released after erythropoietic stimuli to mobilize iron for erythropoiesis[21]. It has been found that erythroblasts obtain mitochondria from central macrophages in stress erythropoiesis through CD47 signaling[22]. It is necessary to further investigate how erythroblast signaling regulates central macrophages in erythropoiesis.

TFPI is identified as a Kunitz-type serine protease inhibitor that endogenously suppresses the tissue factor pathway in coagulation system[23]. TFPI has been implicated in inflammation[24], hematopoietic stem cell homing[25], and vascular development[26]. Accumulating platelets in the growing blood clot release TFPI, firstly named as lipoprotein associated coagulation inhibitor[27]. Furthermore, TFPI is also expressed in endothelial cells[24], fibroblasts[28], and muscle cells[29]. Here, we found that TFPI was highly expressed in erythroblasts, while lowly expressed in central macrophages. Erythroid lineage-specific TFPI deficiency led to reduced erythropoiesis in BM. Moreover, TFPI interacted with thrombomodulin (Thbd) as a functional receptor in central macrophages to promote heme production, which contributed to stress erythropoiesis.

## Results

### TFPI knockout impairs erythropoiesis

To investigate the relevance of blood coagulation to erythropoiesis, we initially assessed plasma TF and TFPI in healthy controls and polycythemia patients with $JAK2^{V617F}$ mutation. Plasma TF was higher in $JAK2^{V617F}$-mutated patients (Fig. 1A), suggesting a potential link between erythropoiesis and coagulation activation. We then established in vitro erythroblastic islands with erythroblasts and macrophages derived from human cord blood cells (Fig. 1B, C), to explore the role of TF and TFPI in erythropoiesis. The results showed that TFPI expression of erythroblasts was upregulated in $JAK2^{V617F}$ mutation, while TF expression of erythroblasts remained unchanged (Fig. 1D). Co-culture with

macrophages did not influence the TFPI levels in erythroblasts (Fig. 1E). Moreover, Gene Expression Commons (GEXC) analysis showed that in BM, TFPI expression was highest in primitive colony-forming-unit erythroid (pCFU-E) among monocyte, megakaryocyte progenitor (MkP), hematopoietic stem cell (HSC), and pCFU-E in mice (supplemental Fig. 1A). The expression of TFPI in erythroid lineages was also up-regulated in $Jak2^{V617F}$-mutated mice (supplemental Fig. 1B). Ly6G/Ter119-F4/80+CD169+Vcam-1+ cells were sorted to represent central macrophage-enriched cells (F4/80+CD169+Vcam-1+ macrophages) in our study[12,22,30]. The expression of TFPI in erythroid lineages was observed to be higher than that in F4/80+CD169+Vcam-1+ macrophages (Fig. 1F). Therefore, we next generated shRNA against TFPI to gain further insights into the role of TFPI in erythropoiesis. TFPI knockdown reduced RBC numbers, hemoglobin (Hb), and hematocrit (HCT) in peripheral blood (PB, supplemental Fig. 1C, D). The stages (referred to here as RI, RII, RIII, RIV, and RV) of erythropoiesis were characterized by flow cytometry using the expression of Ter119, CD71, and CD44. The cell numbers of stages RIII and RIV of erythropoiesis were increased, while the cell number of RV was decreased after shTFPI treatment (supplemental Fig. 1E).

We then employed EpoR-Cre mice to prepare erythroid lineage-specific TFPI knockout mice ($TFPI^{f/f;EpoR}$, Fig. 1G). PB RBC numbers, Hb, and HCT were decreased in $TFPI^{f/f;EpoR}$ mice (Fig. 1H). The cell numbers of stages RIII and RIV of erythropoiesis were increased, while the cell number of RV was decreased in $TFPI^{f/f;EpoR}$ mice (Fig. 1I). Moreover, erythroid lineage-specific TFPI knockout also led to increased apoptosis in erythroblasts (Fig. 1J). The CFU-E colonies were decreased in $TFPI^{f/f;EpoR}$ mice (Fig. 1K). EBI numbers in BM and spleen were also decreased in $TFPI^{f/f;EpoR}$ mice (Fig. 1L, M). To examine the role of TFPI in polycythemia development, we crossed $Vav-iCre;Jak2^{V617F/+}$ mice with $TFPI^{f/f}$ mice and analyzed the resulting double mutant $Jak2^{V617F};TFPI^{f/f;Vav}$ mice. Erythroid lineage-specific TFPI deficiency increased thrombus size and weight but did not influence mice survival rate in $Jak2^{V617F}$ mice (supplemental Fig. 1F, G), consistent with previous study indicating that endothelial cells specific TFPI knockout only resulted in a relatively mild thrombosis phenotype[31]. However, erythroid lineage-specific TFPI deficiency resulted in decreased PB RBC numbers, Hb, and HCT as well as impaired terminal differentiation (Fig. 1N, O). Hypoxia can also induce polycythemia which lead to an increase in erythropoiesis[2]. TFPI expression in erythroblasts of human EBIs was upregulated after exposed to hypoxia (supplemental Fig. 1H). RNA-seq data showed that TFPI was one of the most up-regulated secreted factor genes after 1 week hypoxia exposure (supplemental Fig. 1I). The protein level of TFPI in erythroid lineages was also up-regulated in hypoxia-exposed mice (supplemental Fig. 1J). Both hypoxia-inducible factor (HIF)−1α and HIF-2α exhibit the lowest expression in erythroid lineages among hematopoietic cells based on Haemosphere, a publicly available RNA-seq database (supplemental Fig. 1K, L). PB RBC numbers, Hb, HCT, and CFU-E colonies were increased in hypoxia-exposed mice (supplemental Fig. 1M, N). New RBCs will be produced rapidly under stress conditions[32]. In stress erythropoiesis induced by phenylhydrazine (PHZ), $TFPI^{f/f;EpoR}$ mice developed more severe erythropoietic impairment and had a delayed RBC recovery response compared to $TFPI^{f/f}$ mice (supplemental Fig. 1O, P). The erythropoietic impairment in PHZ-administered $TFPI^{f/f;EpoR}$ mice observed on Day 7 was preceded by an increase in stage RIII and RIV erythroblast frequency on Day 6 (supplemental Fig. 1Q). The results showed that TFPI regulated erythropoiesis under both steady state and stress conditions.

### Macrophages involved in the effect of TFPI on erythropoiesis

To determine whether TFPI acts directly on erythroblasts, we measured the proliferation of Ter119+CD71+ cells sorted from $TFPI^{f/f}$ mice and $TFPI^{f/f;EpoR}$ mice (Fig. 2A, supplemental Fig. 2A). However, both TFPI knockout and TFPI shRNA treatment failed to induce changes of in vitro cultured erythroblast population proliferation in

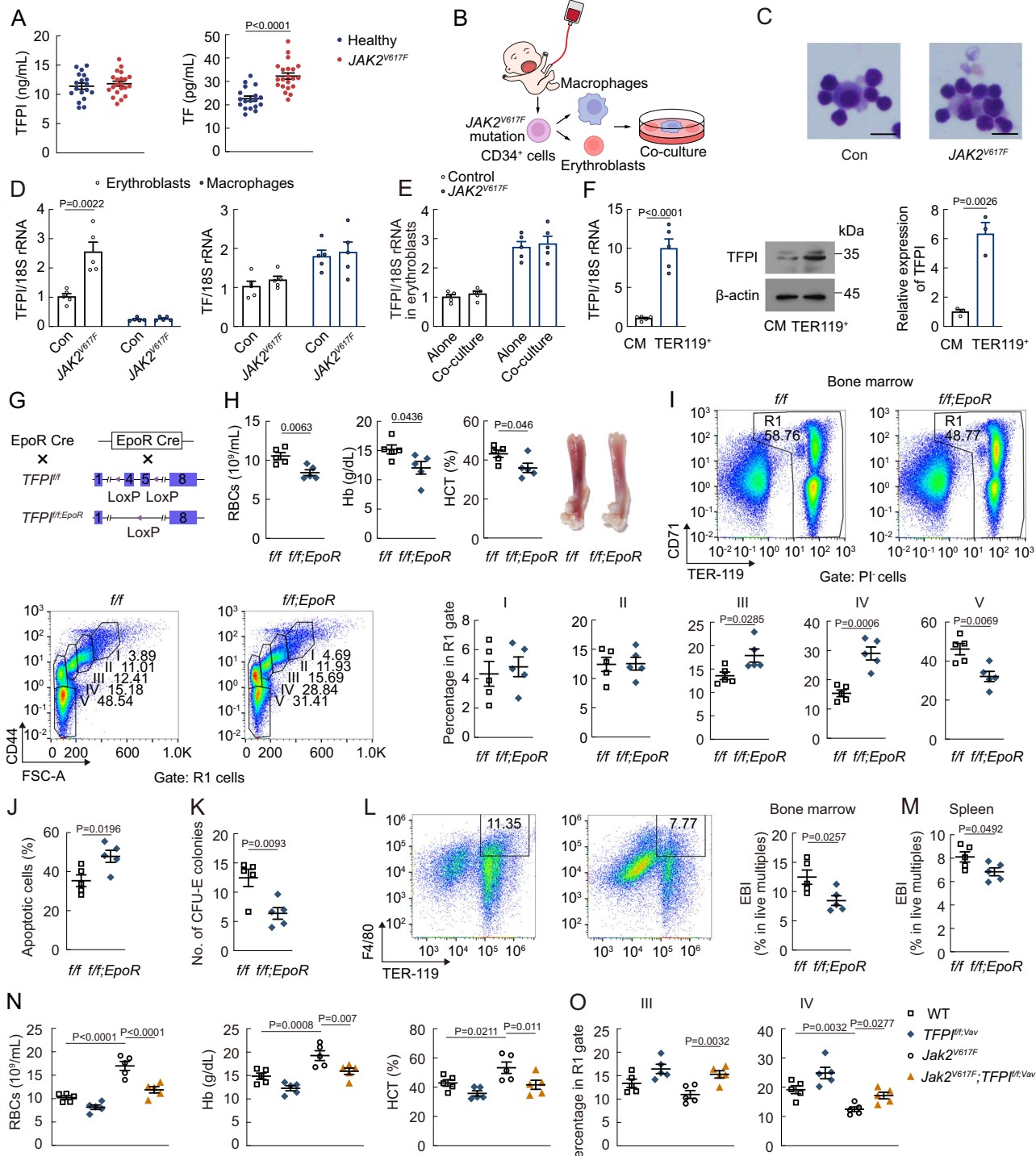

**Fig. 1 | Erythroid lineage-specific TFPI knockout results in erythropoietic impairment in mice. A** TFPI and TF concentrations in the plasma of healthy controls and *JAK2V617F* mutation polycythemia (Healthy, *n* = 18; *JAK2V617F*, *n* = 21). **B** Protocol used for human EBI formation. **C** Representative cytospin images of EBIs formed between erythroblasts and macrophages with *JAK2V617F* mutation (*n* = 3). Scale bars, 10 µm. **D** TFPI and TF mRNA expression in erythroblasts or macrophages with *JAK2V617F* mutation (*n* = 5). **E** TFPI mRNA expression in erythroblasts cultured with or without macrophages. Alone, erythroblasts culture independently; co-culture, erythroblasts co-culture with macrophages (*n* = 5). **F** TFPI mRNA and protein expression in F4/80⁺CD169⁺Vcam-1⁺ macrophages (CM) and Ter119⁺ cells (RT-qPCR, *n* = 5; Western blot, *n* = 3). **G** Protocol used to prepare *TFPI*ᶠ/ᶠ;ᴱᵖᵒᴿ mice. **H** PB RBC

numbers, Hb, and HCT in *TFPI*ᶠ/ᶠ;ᴱᵖᵒᴿ mice (*n* = 5). **I** Frequency of erythroblast populations among BM cells in *TFPI*ᶠ/ᶠ;ᴱᵖᵒᴿ mice (*n* = 5). **J** Frequency of apoptotic erythroblasts among BM cells in *TFPI*ᶠ/ᶠ;ᴱᵖᵒᴿ mice (*n* = 5). **K** CFU-E number in *TFPI*ᶠ/ᶠ;ᴱᵖᵒᴿ mice. (*n* = 5). **L** EBI (F4/80⁺Ter119⁺ live multiplets) numbers in the BM of *TFPI*ᶠ/ᶠ;ᴱᵖᵒᴿ mice. (*n* = 5). **M** EBI numbers in the spleen of *TFPI*ᶠ/ᶠ;ᴱᵖᵒᴿ mice (*n* = 5). **N** PB RBC numbers, Hb, and HCT in *Jak2V617F*-mutated, *TFPI*ᶠ/ᶠ;ⱽᵃᵛ, or *Jak2V617F*;*TFPI*ᶠ/ᶠ;ⱽᵃᵛ mice (*n* = 5). **O** Frequency of RIII and RIV erythroblast populations among BM cells in *Jak2V617F*-mutated, *TFPI*ᶠ/ᶠ;ⱽᵃᵛ, or *Jak2V617F*; *TFPI*ᶠ/ᶠ;ⱽᵃᵛ mice (*n* = 5). Statistical analysis was performed using one-way ANOVA (**A**, **D**, **F**, **H–I**, **J–K** and **L–O**). Data are shown as mean ± SEM and are representative of two (**C**, **H–I**, and **L–O**) or three (**A**, **D–F**, and **J**, **K**) independent experiments. Source data are provided as a Source Data file.

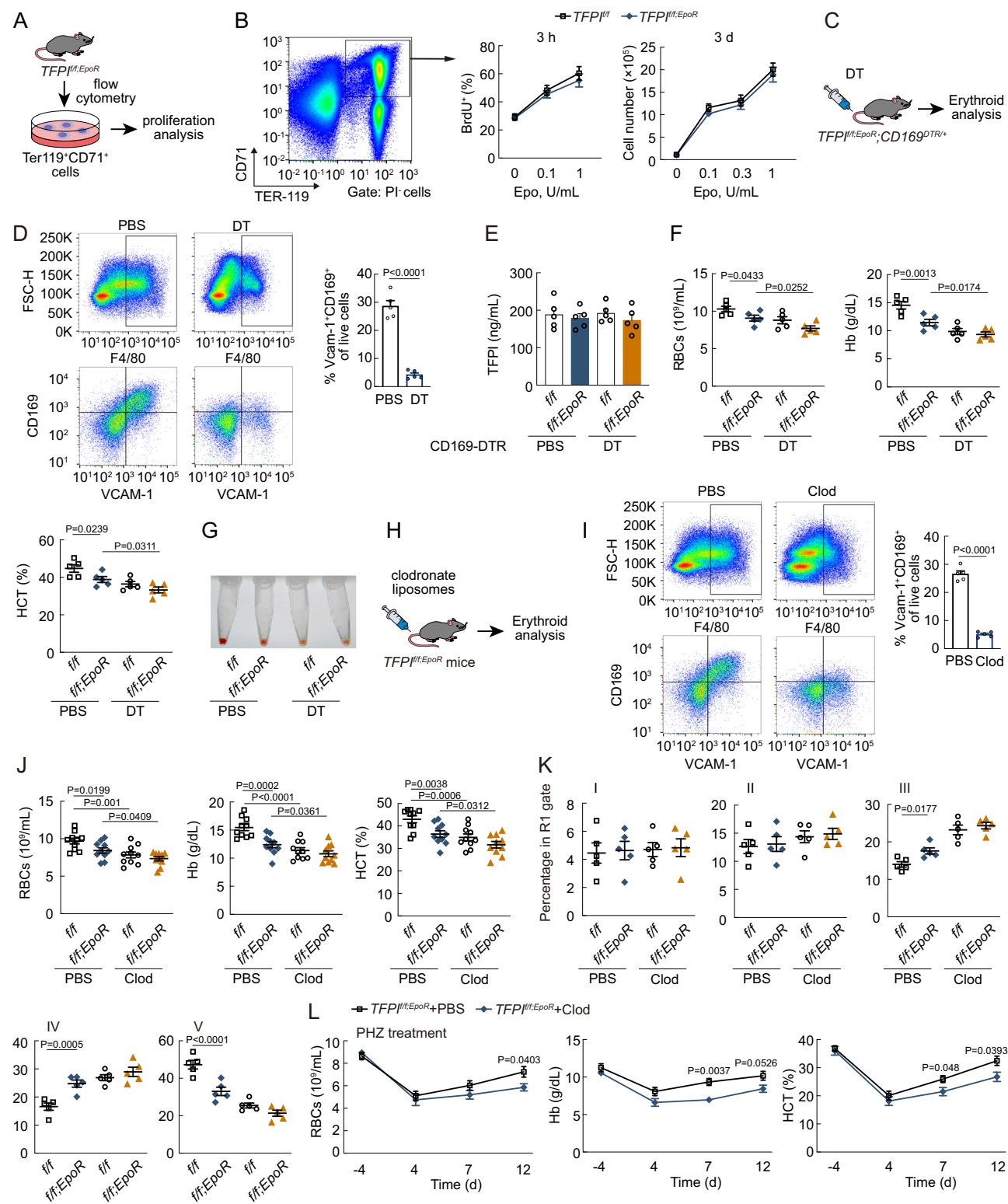

**Fig. 2 | Macrophages involved in the effect of TFPI on erythropoiesis.**
**A** Experimental design to determine the effect of TFPI on the proliferation of Ter119⁺CD71⁺ cells in vitro. **B** The effect of erythroid lineage-specific TFPI knockout on the expansion of Ter119⁺CD71⁺ cells (*n* = 5). **C** Schematic diagram of double mutant *TFPI^f/f;EpoR;CD169^DTR/+* mice. **D** FACS plots of F4/80⁺CD169⁺Vcam-1⁺ macrophages of *CD169^DTR/+* mice after DT treatment (*n* = 5). **E** The plasma TFPI concentration of *TFPI^f/f;EpoR;CD169^DTR/+* mice (*n* = 5). **F** PB RBC numbers, Hb, and HCT in *TFPI^f/f;EpoR;CD169^DTR/+* mice after DT treatment (*n* = 5). **G** BM cell pellets from *TFPI^f/f;EpoR;CD169^DTR/+* mice after DT treatment. **H** Schematic diagram of macrophage depletion using clodronate liposomes in *TFPI^f/f;EpoR* mice. **I** FACS plots of F4/

80⁺CD169⁺Vcam-1⁺ macrophages of *TFPI^f/f;EpoR* mice after clodronate liposomes treatment (*n* = 5). **J** PB RBC numbers, Hb, and HCT in *TFPI^f/f;EpoR* mice after clodronate liposomes treatment (*n* = 10). **K** Frequency of erythroblast populations among BM cells *TFPI^f/f;EpoR* mice after clodronate liposomes treatment (*n* = 5). **L** PB RBC numbers, Hb, and HCT of *TFPI^f/f;EpoR* mice after clodronate liposomes and PHZ treatment (*n* = 5). Statistical analysis was performed using one-way ANOVA (**D**, **F**, **I**, **J**–**L**). Data are shown as mean ± SEM and are representative of two (**D**, **F**, **G**, and **I**–**L**) or three (**B** and **E**) independent experiments. Source data are provided as a Source Data file.

bromodeoxyuridine (BrdU) incorporation experiments (Fig. 2B, supplemental Fig. 2B, C). Furthermore, the proliferation of erythroblasts treated with TFPI shRNA was reduced when co-cultured with macrophages (supplemental Fig. 2D, E), indicating the potential involvement of macrophages in the effect of TFPI on erythropoiesis. Consequently, we further employed double mutant $TFPI^{f/f;EpoR};CD169^{DTR/+}$ mice to investigate whether macrophages mediate the effect of TFPI on erythropoiesis in vivo (Fig. 2C). F4/80+CD169+Vcam-1+ macrophages in $CD169^{DTR/+}$ mice were depleted after diphtheria toxin (DT) treatment (Fig. 2D). The plasma TFPI concentration remained unchanged after erythroid lineage-specific TFPI knockout (Fig. 2E). Erythroid lineage-specific TFPI knockout resulted in a decrease in PB RBC numbers, Hb, and HCT, but failed to induce further decrease in erythropoiesis on the $CD169^{DTR/+}$ background after DT treatment (Fig. 2F, G). We further used clodronate liposomes to deplete macrophages (Fig. 2H, I). Erythroid lineage-specific TFPI knockout did not influence erythropoiesis in mice after clodronate liposomes treatment as well (Fig. 2J). Moreover, erythroid lineage-specific TFPI knockout disrupted terminal differentiation of erythroblasts, resulting in the block of erythroblast differentiation at III and IV stages (Fig. 2K). However, in clodronate liposome-treated groups, the cell numbers of stages RIII, RIV, and RV in BM showed no substantial change in $TFPI^{f/f;EpoR}$ mice (Fig. 2K). Additionally, macrophage depletion itself affected erythropoiesis both in $TFPI^{f/f}$ and $TFPI^{f/f;EpoR}$ mice (Fig. 2J). In stress erythropoiesis induced by PHZ, macrophage depletion also caused a delayed RBC recovery response in $TFPI^{f/f;EpoR}$ mice (Fig. 2L). Above all, the results illustrated that the effect of TFPI on erythropoiesis was mediated by macrophages.

We injected rTFPI to further study the effect of TFPI on macrophage-mediated erythropoiesis. The plasma TFPI concentration was higher in rTFPI-treated mice (supplemental Fig. 2F). Moreover, rTFPI treatment increased PB RBC numbers, Hb, and HCT and promoted terminal differentiation in both BM and spleen (supplemental Fig. 2G-I). rTFPI treatment also increased EBI numbers in BM and spleen (supplemental Fig. 2J, K). In PHZ-induced stress erythropoiesis model, rTFPI treatment also resulted in increased PB RBC numbers, Hb, and HCT as expected (supplemental Fig. 2L, M). However, rTFPI treatment did not result in an increase in PB RBC numbers, Hb, and HCT, nor did it influence terminal differentiation in $CD169^{DTR/+}$ mice after DT treatment (supplemental Fig. 2N, O).

Since F4/80+CD169+Vcam-1+ macrophages also have slight TFPI expression (Fig. 1F), we employed macrophage-specific TFPI knockout mice to further explore the effect of macrophage-derived TFPI on erythropoiesis (supplemental Fig. 3A). TFPI expression was ablated completely in F4/80+CD169+Vcam-1+ macrophages of $TFPI^{f/f;CD169}$ mice (supplemental Fig. 3B). We did not observe significant changes in PB RBC numbers, Hb, and HCT in $TFPI^{f/f;CD169}$ mice (supplemental Fig. 3C). $TFPI^{f/f;CD169}$ mice also showed no difference in cell numbers of stages RI to RV of erythropoiesis (supplemental Fig. 3D). Furthermore, macrophage-specific TFPI knockout did not induce changes in erythroblast apoptosis and proliferation (supplemental Fig. 3E, F). Additionally, the mRNA levels of genes required for mature RBC production including HBA-A1, HBB-B1, GYPA, EPB41, and AQP1 in erythroblasts also remained unchanged (supplemental Fig. 3G). Therefore, erythroid lineage-derived, but not macrophage-derived, TFPI promoted erythropoiesis in bone marrow.

## TFPI increases heme production in F4/80+CD169+Vcam-1+ macrophages

F4/80+CD169+Vcam-1+ macrophages were isolated from $TFPI^{f/f}$ or $TFPI^{f/f;EpoR}$ mice (supplemental Fig. 4A), and RNA sequencing (RNA-seq) was performed to understand the molecular mechanism underlying the impairment of erythropoiesis following TFPI knockout. Analysis of RNA-seq data by Gene Ontology (GO) of differentially expressed genes showed that erythroid lineage-specific ablation of TFPI denoted down-

regulation of several pathways in F4/80+CD169+Vcam-1+ macrophages including heme biosynthetic process and porphyrin-containing compound biosynthetic process, which were related to heme production (Fig. 3A, B). RNA-seq data revealed the down-regulation of seven enzymes in heme biosynthesis including 5-aminolevulinate synthase 2 (ALAS2), Delta-aminolevulinic acid dehydratase (ALAD), hydroxymethylbilane synthase (HMBS), uroporphyrinogen III synthase (UROS), uroporphyrinogen decarboxylase (UROD), coproporphyrinogen III oxidase (CPOX), and Ferrochelatase (Fech) at the mRNA level (Fig. 3B). Fech is the rate-limiting enzyme that incorporates ferrous iron into protoporphyrin IX (PPIX) in heme biosynthesis in final step[33]. We found that the mRNA levels of Fech were significantly reduced in F4/80+CD169+Vcam-1+ macrophages of $TFPI^{f/f;EpoR}$ mice but increased after rTFPI treatment (Fig. 3C, D). Fech mRNA level was decreased in F4/80+CD169+Vcam-1+ macrophages but did not change in Ter119+ cells of $TFPI^{f/f;EpoR}$ mice (Fig. 3E). $TFPI^{f/f;EpoR}$ mice showed a reduction in heme content in Ter119+ cells and F4/80+CD169+Vcam-1+ macrophages compared with $TFPI^{f/f}$ mice (Fig. 3F, G). These findings supported the fact that erythroid lineage-specific TFPI knockout impaired heme synthesis in F4/80+CD169+Vcam-1+ macrophages. Moreover, Heme content of Ter119+ cells after macrophage-depletion was similar in $TFPI^{f/f;EpoR}$ and $TFPI^{f/f}$ mice (Fig. 3H, I). GATA1 has been reported as an essential regulator of erythroid cell gene expression[34,35], serving as a direct transcription factor for Fech[36]. We found that the phosphorylation level of GATA1 in F4/80+CD169+Vcam-1+ macrophages of $TFPI^{f/f;EpoR}$ mice increased over time after rTFPI treatment (Fig. 3J). Additionally, rTFPI treatment did not change the heme content in F4/80+CD169+Vcam-1+ macrophages after GATA1 shRNA treatment (Fig. 3K, L). These data suggested that erythroid lineage-specific TFPI knockout impaired heme biosynthesis in F4/80+CD169+Vcam-1+ macrophages.

To further investigate the role of heme in F4/80+CD169+Vcam-1+ macrophages during erythropoiesis, we first found the mRNA and protein levels of Fech in F4/80+CD169+Vcam-1+ macrophages increased in $Jak2^{V617F}$-mutated, hypoxia-exposed, and PHZ-treated mice (supplemental Fig. 4B–D). We then employed CD169-Cre mice to prepare macrophage-specific Fech knockout mice ($Fech^{f/f;CD169}$, supplemental Fig. 4E), which showed a decrease in F4/80+CD169+Vcam-1+ macrophage heme content, PB RBC numbers, Hb, and HCT (supplemental Fig. 4F, G). Furthermore, macrophage-specific Fech knockout prevented terminal differentiation and promoted apoptosis in erythroblasts (supplemental Fig. 4H, I). After PHZ treatment, $Fech^{f/f;CD169}$ mice developed more severe erythropoietic impairment and had a delayed RBC recovery response (supplemental Fig. 4J). Additionally, rTFPI treatment failed to increase F4/80+CD169+Vcam-1+ macrophage heme content, PB RBC numbers, Hb, and HCT in $Fech^{f/f;CD169}$ mice (supplemental Fig. 4K, L), suggesting that TFPI promoted erythropoiesis by regulating Fech expression in F4/80+CD169+Vcam-1+ macrophages.

## TFPI interacts with Thbd in F4/80+CD169+Vcam-1+ macrophages

To understand the mechanism by which TFPI promotes erythropoiesis, we aimed to identify a potential TFPI receptor. Initially, we investigated whether TF played a role in the function of TFPI in erythropoiesis. However, no alterations were observed in TF expression in F4/80+CD169+Vcam-1+ macrophages of $Jak2^{V617F}$-mutated or hypoxia-exposed mice (Fig. 4A). We blocked TF activity by employing a monoclonal antibody, which reduced TF procoagulant activity (PCA) and plasma levels of thrombin-antithrombin (TAT, Fig. 4B). Subsequent TF monoclonal antibody (mAb) treatment in both $TFPI^{f/f}$ and $TFPI^{f/f;EpoR}$ mice failed to induce changes in heme content and Fech expression in F4/80+CD169+Vcam-1+ macrophages as well as erythropoiesis (Fig. 4C–E). We then carried out a yeast two-hybrid screen to search for candidate TFPI-interacting proteins (Table S1), and confirmed the interaction between TFPI and Thbd by co-immunoprecipitation (Co-IP, Fig. 4F). To narrow down the region of

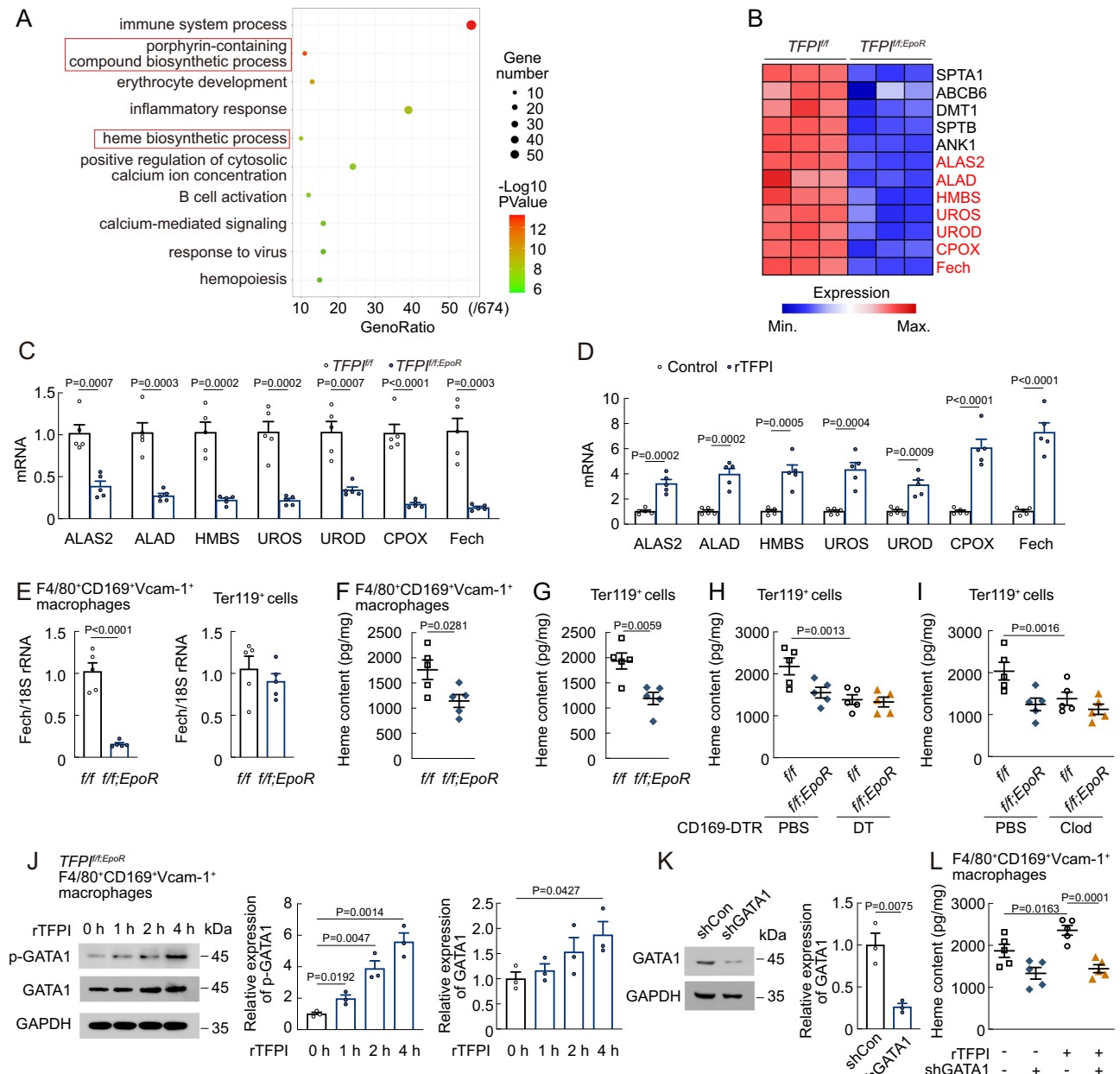

**Fig. 3 | TFPI affects signaling pathways in F4/80+CD169+Vcam-1+ macrophages.**
**A** GO term enrichment analysis of differentially expressed genes in F4/80+CD169+Vcam-1+ macrophages between *TFPI^f/f* and *TFPI^f/f;EpoR* mice. **B** Heatmap of differential gene expression. **C** mRNA expression of ALAS2, ALAD, HMBS, UROS, UROD, CPOX, and Fech in F4/80+CD169+Vcam-1+ macrophages of *TFPI^f/f;EpoR* mice (*n* = 5). **D** mRNA expression of ALAS2, ALAD, HMBS, UROS, UROD, CPOX, and Fech in F4/80+CD169+Vcam-1+ macrophages of rTFPI-treated mice (*n* = 5). **E** Fech mRNA expression in F4/80+CD169+Vcam-1+ macrophages and Ter119+ cells of *TFPI^f/f;EpoR* mice (*n* = 5). (F and G) Heme content in F4/80+CD169+Vcam-1+ macrophages **F** and Ter119+ cells **G** of *TFPI^f/f;EpoR* mice (*n* = 5). **H** Heme content in Ter119+ cells of *TFPI^f/*

*f;EpoR*;*CD169^DTR/+* mice after DT treatment (*n* = 5). **I** Heme content in Ter119+ cells of *TFPI^f/f;EpoR* mice after clodronate liposomes treatment (*n* = 5). **J** p-GATA1 and GATA1 protein levels in F4/80+CD169+Vcam-1+ macrophages of *TFPI^f/f;EpoR* mice (*n* = 3). **K** GATA1 protein expression of F4/80+CD169+Vcam-1+ macrophages after GATA1 shRNA treatment (*n* = 3). **L** Heme content in F4/80+CD169+Vcam-1+ macrophages of *TFPI^f/f* mice after treated with rTFPI and GATA1 shRNA (*n* = 5). Statistical analysis was performed using one-way ANOVA (**A**, and **C–L**). Data are shown as mean ± SEM and are representative of two (**E–I**) or three (C, D, and J-L) independent experiments. Source data are provided as a Source Data file.

Thbd that mediated its binding to TFPI, we divided the extracellular region of Thbd into two segments based on its structure. The Thbd (205-518) segment, but not Thbd (17-204) segment, interacted with TFPI (Fig. 4G, H). We also confirmed that TFPI interacted with Thbd in F4/80+CD169+Vcam-1+ macrophages (Fig. 4I). Additionally, GST pull-down assays showed that Thbd interacted with TFPI (Fig. 4J). We next sought to determine whether Thbd was involved in TFPI-mediated promotion of heme production. The mRNA level of Fech was not increased by rTFPI treatment after Thbd shRNA treatment in F4/

80+CD169+Vcam-1+ macrophages (Fig. 4K, L), but it was increased by rTFPI treatment after Thbd transfection in HEK293T cells (Fig. 4M).

TFPI-2 is another secreted factor gene homologous to TFPI, so we further investigated the interaction between TFPI-2 and Thbd. Sequence alignment showed that TFPI-2 shared low identity with TFPI (supplemental Fig. 5A). We noted a lower TFPI-2 content in BM than in plasma (supplemental Fig. 5B). Additionally, our results showed that TFPI-2 did not interact with Thbd (supplemental Fig. 5C), indicating that Thbd specifically interacted with TFPI.

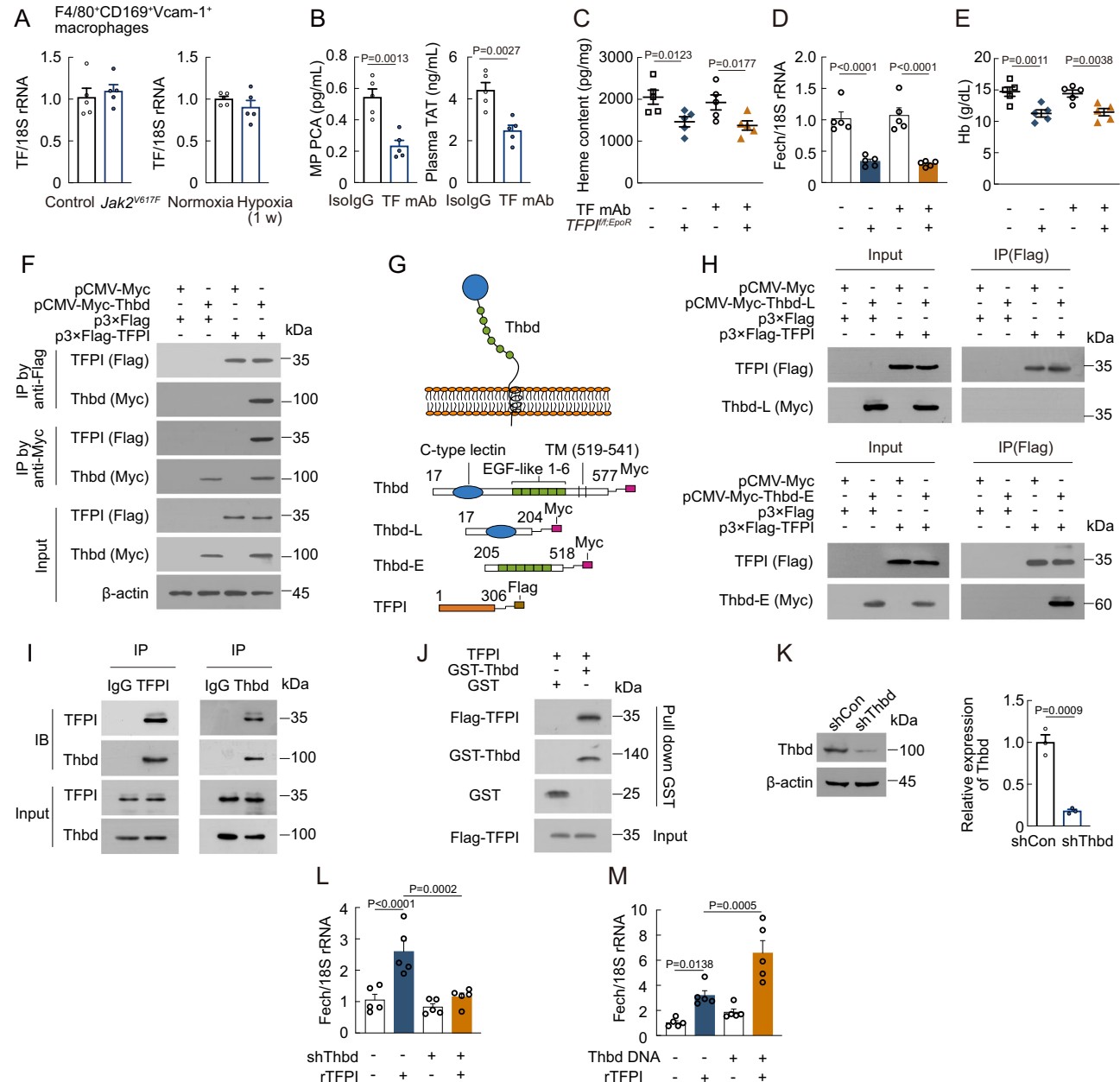

**Fig. 4 | TFPI interacts with Thbd in F4/80+CD169+Vcam-1+ macrophages.**
**A** mRNA expression of TF in F4/80+CD169+Vcam-1+ macrophages of *Jak2V617F*-mutated or hypoxia-exposed mice (*n* = 5). **B** TF procoagulant activity and plasma TAT levels in mice after TF mAb treatment (*n* = 5). **C** Heme content in F4/80+CD169+Vcam-1+ macrophages of *TFPIf/f;EpoR* mice after TF mAb treatment (*n* = 5). **D** Fech mRNA expression in F4/80+CD169+Vcam-1+ macrophages of *TFPIf/f;EpoR* mice after TF mAb treatment (*n* = 5). **E** PB Hb in *TFPIf/f;EpoR* mice after TF mAb treatment (*n* = 5). **F** Co-IP analysis of the interaction between TFPI and Thbd in HEK293T cells (*n* = 2). **G** Schematic illustration of Thbd and TFPI constructs. **H** Co-IP analysis of the interaction of TFPI with the different domains of Thbd in HEK293T cells (*n* = 2). **I** Co-

IP analysis of the interaction between TFPI and Thbd in F4/80+CD169+Vcam-1+ macrophages (*n* = 2). **J** Pull down analysis of the interaction between the TFPI and Thbd (*n* = 2). **K** Thbd protein expression in F4/80+CD169+Vcam-1+ macrophages after Thbd shRNA treatment (*n* = 3). **L** Fech mRNA expression in F4/80+CD169+Vcam-1+ macrophages after Thbd shRNA and rTFPI treatment (*n* = 5). **M** Fech mRNA expression in HEK293T cells after Thbd transfection and rTFPI treatment (*n* = 5). Statistical analysis was performed using one-way ANOVA (**B**–**E** and **K**–**M**). Data are shown as mean ± SEM and are representative of two (**B**–**J**) or three (**A** and **K**–**M**) independent experiments. Source data are provided as a Source Data file.

## Macrophage-specific Thbd knockout decreases erythropoiesis
Since CD169-cre transgene and Thbd gene are located very close on chromosome 2, *Thbdf/f;LysM* mice were prepared to explore the effect of macrophage-specific Thbd knockout on erythropoiesis (Fig. 5A). Our findings revealed that Fech expression levels and heme content were reduced in F4/80+CD169+Vcam-1+ macrophages of *Thbdf/f;LysM* mice (Fig. 5B, C), which also resulted in decreased PB RBC numbers, Hb, and HCT (Fig. 5D). Additionally, macrophage-specific Thbd knockout prevented terminal differentiation, promoted apoptosis in

erythroblasts and decreased EBI numbers in BM and spleen (Fig. 5E-5H). After PHZ treatment, *Thbdf/f;LysM* mice showed a delayed RBC recovery response and impaired terminal differentiation (Fig. 5I, J). Furthermore, rTFPI treatment failed to increase F4/80+CD169+Vcam-1+ macrophage heme content, as well as PB RBC numbers, Hb, and HCT in *Thbdf/f;LysM* mice (Fig. 5K, L). The results suggested that Thbd in F4/80+CD169+Vcam-1+ macrophages were required for erythropoiesis.

We next employed erythroid lineage-specific Thbd knockout mice to determine whether erythroid-expressed Thbd also regulated

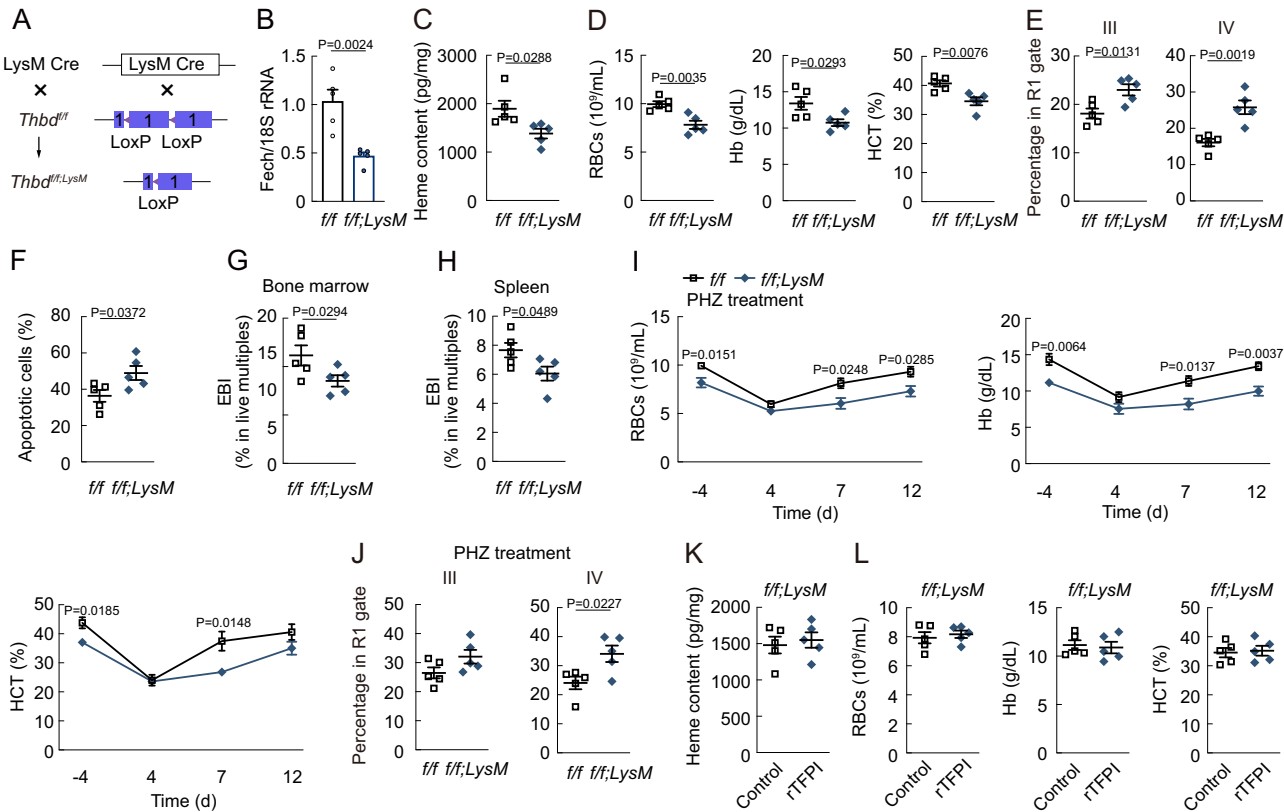

**Fig. 5 | Macrophage-specific Thbd knockout decreases erythropoiesis.**
**A** Protocol to prepare *Thbd^f/f;LysM* mice. **B** Fech expression in F4/80⁺CD169⁺Vcam-1⁺ macrophages of *Thbd^f/f;LysM* mice (*n* = 5). **C** Heme content in F4/80⁺CD169⁺Vcam-1⁺ macrophages of *Thbd^f/f;LysM* mice (*n* = 5). **D** PB RBC numbers, Hb, and HCT of *Thbd^f/f;LysM* mice (*n* = 5). **E** Frequency of RIII and RIV erythroblast populations in *Thbd^f/f;LysM* mice (*n* = 5). **F** Frequency of apoptotic erythroblasts among BM cells in *Thbd^f/f;LysM* mice (*n* = 5). **G** EBI numbers in the BM of *Thbd^f/f;LysM* mice (*n* = 5). **H** EBI numbers in the spleen of *Thbd^f/f;LysM* mice (*n* = 5). **I** PB RBC numbers, Hb, and HCT of *Thbd^f/f;LysM*

mice after PHZ treatment (*n* = 5). **J** Frequency of RIII and RIV erythroblast populations in *Thbd^f/f;LysM* mice 6 d after PHZ treatment (*n* = 5). **K** Heme content in F4/80⁺CD169⁺Vcam-1⁺ macrophages of *Thbd^f/f;LysM* mice after rTFPI treatment (*n* = 5). **L** PB RBC numbers, Hb, and HCT in *Thbd^f/f;LysM* mice after rTFPI treatment (*n* = 5). Statistical analysis was performed using one-way ANOVA (B-J). Data are shown as mean ± SEM and are representative of two (C-L) or three **B** independent experiments. Source data are provided as a Source Data file.

erythropoiesis (supplemental Fig. 6A). However, in *Thbd^f/f;EpoR* mice, no changes were observed in PB RBC numbers, Hb, and HCT, as well as terminal differentiation (supplemental Fig. 6B, C). Heme content of Ter119⁺ cells also remained unchanged in *Thbd^f/f;EpoR* mice (supplemental Fig. 6D). Moreover, erythroid lineage-specific Thbd knockout had no effect on erythroblast apoptosis and proliferation (supplemental Fig. 6E, F). The mRNA levels of HBA-A1, HBB-B1, GYPA, EPB41, and AQP1 in erythroblasts also remained unchanged in *Thbd^f/f;EpoR* mice (supplemental Fig. 6G). The results suggested that erythroid lineage-specific Thbd knockout had no effect on erythropoiesis.

## Thbd promotes the signaling pathway of heme synthesis in F4/80⁺CD169⁺Vcam-1⁺ macrophages
We then examined the downstream signaling pathways of the TFPI/Thbd axis. Previous studies have shown that activated protein C (aPC) plays a crucial role in Thbd signaling[37]. We found that the mRNA levels of aPC in F4/80⁺CD169⁺Vcam-1⁺ macrophages remained unchanged in *Jak2^V617F*-mutated and hypoxia-exposed mice (Fig. 6A). To explore the potential involvement of aPC in the influence of the TFPI/Thbd axis on erythropoiesis, we knocked down aPC in F4/80⁺CD169⁺Vcam-1⁺ macrophages through intraosseous infusion of lentivirus-expressing aPC shRNA (Fig. 6B). aPC shRNA treatment blocked the terminal differentiation of erythroblasts in *Thbd^f/f* mice, but did not in *Thbd^f/f;LysM* mice (Fig. 6C), indicating that aPC mediated the effect of Thbd on erythropoiesis. We further analyzed RNA-seq data of F4/80⁺CD169⁺Vcam-1⁺ macrophages in *TFPI^f/f* and *TFPI^f/f;EpoR* mice. Kyoto Encyclopedia of Genes and Genomes (KEGG) pathway enrichment analysis showed the top 15 down-regulated signaling pathways in *TFPI^f/f;EpoR* mice including FoxO signaling pathway and MAPK signaling pathway (Fig. 6D). We observed a decrease in heme content in F4/80⁺CD169⁺Vcam-1⁺ macrophages after treated with rTFPI combined with an inhibitor of ERK1/2, SCH772984 (Fig. 6E), while the heme content in F4/80⁺CD169⁺Vcam-1⁺ macrophages was not changed after treatment with rTFPI combined with an inhibitor of JNK, JNK-IN-8, and an inhibitor of FOXO1, AS1842856 (Fig. 6E). SCH772984 treatment resulted in no further reduction in heme content after GATA1 shRNA treatment (Fig. 6F), indicating that ERK1/2 participates in GATA1 activation after rTFPI treatment. Moreover, Western blot analysis confirmed the up-regulation of p-ERK1/2 in F4/80⁺CD169⁺Vcam-1⁺ macrophages after rTFPI treatment (Fig. 6G). We then found that the protein level of p-GATA1 was decreased in *Thbd^f/f;LysM* or SCH772984-treated F4/80⁺CD169⁺Vcam-1⁺ macrophages and could not be rescued by rTFPI treatment (Fig. 6H, I). Similarly, protein levels of p-GATA1 were increased over time in F4/80⁺CD169⁺Vcam-1⁺ macrophages of *Thbd^f/f* mice, but remained unchanged in F4/80⁺CD169⁺Vcam-1⁺ macrophages of *Thbd^f/f;LysM* mice after rTFPI treatment (Fig. 6J). Moreover, treatment with rTFPI increased ALAS2 and Fech mRNA levels in F4/80⁺CD169⁺Vcam-1⁺ macrophages of *Thbd^f/f* mice, but not in F4/80⁺CD169⁺Vcam-1⁺ macrophages of *Thbd^f/f;LysM* mice, and this effect was attenuated by SCH772984 and GATA1 shRNA treatment (Fig. 6K−M). Therefore, the results suggested that the TFPI/Thbd axis promoted heme synthesis via the aPC/ERK1/2/GATA1 signaling pathway (Fig. 6N).

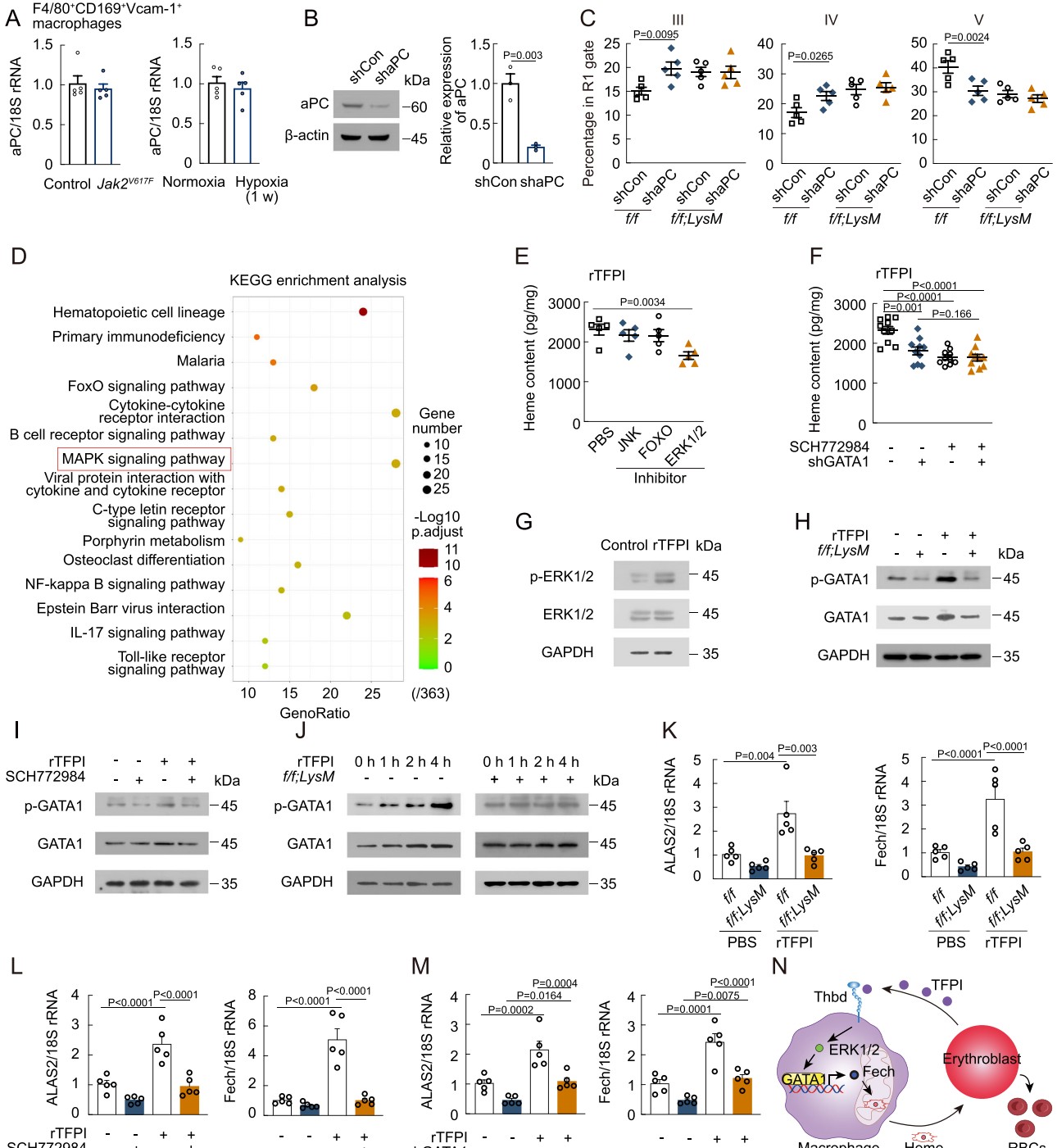

**Fig. 6 | Thbd promotes the signaling pathway of heme synthesis in F4/80⁺CD169⁺Vcam-1⁺ macrophages. A** mRNA expression of aPC in F4/80⁺CD169⁺Vcam-1⁺ macrophages of *Jak2^{V617F}*-mutated and hypoxia-exposed mice (*n* = 5). **B** aPC protein expression of F4/80⁺CD169⁺Vcam-1⁺ macrophages after intraosseous infusion of lentivirus-expressing aPC shRNA (*n* = 3). **C** Frequency of RIII, RIV, and RV erythroblast populations in *Thbd^{f/f;LysM}* mice after intraosseous infusion of lentivirus-expressing aPC shRNA (*n* = 5). **D** KEGG analysis of down-regulated genes in F4/80⁺CD169⁺Vcam-1⁺ macrophages in *TFPI^{f/f;EpoR}* mice. **E** Heme content in F4/80⁺CD169⁺Vcam-1⁺ macrophages treated with rTFPI combined with JNK, FOXO, and ERK1/2 inhibitors (*n* = 5). **F** Heme content in F4/80⁺CD169⁺Vcam-1⁺ macrophages after treated with rTFPI combined with ERK1/2 inhibitor and GATA1 shRNA (*n* = 10). **G** Phosphorylation level of ERK1/2 protein in F4/80⁺CD169⁺Vcam-1⁺ macrophages after rTFPI treatment (*n* = 3). **H** Phosphorylation level of GATA1 protein in F4/80⁺CD169⁺Vcam-1⁺ macrophages of *Thbd^{f/f;LysM}* mice

after rTFPI treatment (*n* = 3). **I** Phosphorylation level of GATA1 protein in F4/80⁺CD169⁺Vcam-1⁺ macrophages treated with rTFPI and ERK1/2 inhibitor (*n* = 3). **J** Phosphorylation level of GATA1 protein in F4/80⁺CD169⁺Vcam-1⁺ macrophages of *Thbd^{f/f;LysM}* mice at different time points after rTFPI treatment (*n* = 3). **K** ALAS2 and Fech mRNA expression in F4/80⁺CD169⁺Vcam-1⁺ macrophages of *Thbd^{f/f;LysM}* mice after rTFPI treatment (*n* = 5). **L** ALAS2 and Fech mRNA expression in F4/80⁺CD169⁺Vcam-1⁺ macrophages after treated with rTFPI and ERK1/2 inhibitor (*n* = 5). **M** ALAS2 and Fech mRNA expression in F4/80⁺CD169⁺Vcam-1⁺ macrophages after treated with rTFPI and GATA1 shRNA (*n* = 5). Statistical analysis was performed using one-way ANOVA (**B**, **C**, **E**, **F**, and **K–M**). Data are shown as mean ± SEM and are representative of two (**C**, **E**, and **F**) or three (**A**, **B**, and **G–M**) independent experiments. **N** The signaling pathway of TFPI/Thbd mediated heme synthesis in F4/80⁺CD169⁺Vcam-1⁺ macrophages. Source data are provided as a Source Data file.

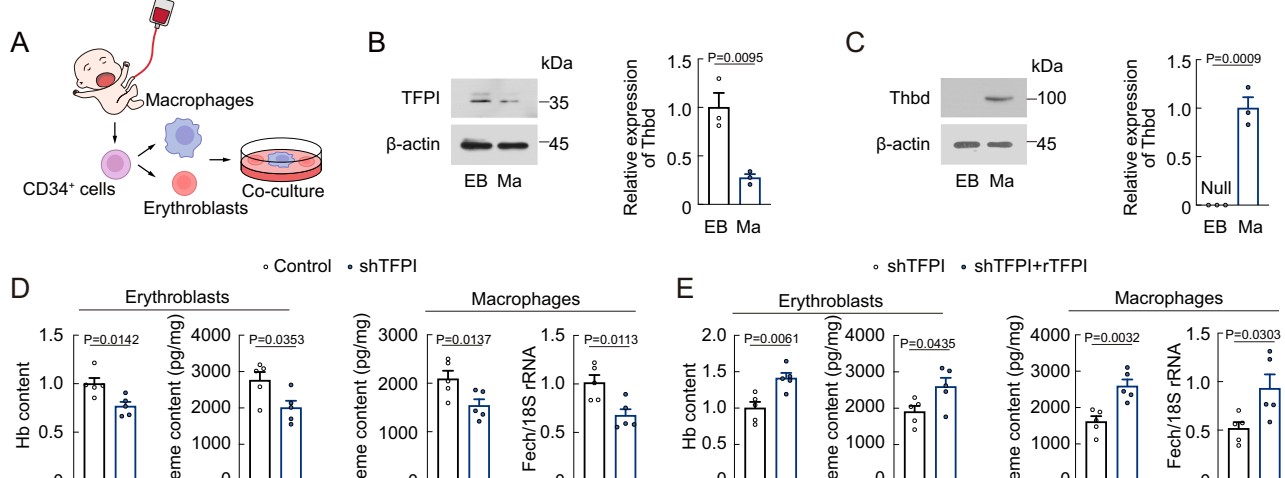

**Fig. 7 | TFPI contributes to human EBI formation and erythropoiesis. A** Protocol used for human EBI formation. **B** TFPI expression in erythroblasts (EB) and macrophages (Ma) ($n = 3$). **C** Thbd expression in erythroblasts and macrophages. Null means no expression ($n = 3$). **D** hemoglobin and heme content in erythroblasts, heme content and Fech mRNA expression in macrophages of EBIs formed by macrophages and erythroblasts treated with TFPI shRNA ($n = 5$). **E** hemoglobin and heme content in erythroblasts, heme content and Fech mRNA expression in macrophages of EBIs formed by macrophages and erythroblasts treated with TFPI shRNA and rTFPI ($n = 5$). Statistical analysis was performed using one-way ANOVA (**B–E**). Data are shown as mean ± SEM and are representative of two (D and E) or three (B and C) independent experiments. Source data are provided as a Source Data file.

## TFPI knockdown represses erythropoiesis in polycythemia

Since our results indicated that elevated levels of TFPI led to increased erythropoiesis, we then performed human EBI formation assay using macrophages and erythroblasts derived from human cord blood CD34+ cells (Fig. 7A), and examined the expression of TFPI and Thbd in erythroblasts and macrophages. The TFPI protein level was higher in erythroblasts, while the Thbd protein level was higher in macrophages (Fig. 7B, C). To further explore the roles of TFPI on human erythropoiesis, we pretreated erythroblasts with TFPI shRNA before co-culturing with macrophages. The Hb and heme content in erythroblasts decreased after TFPI shRNA treatment, while the Fech mRNA expression and heme content in macrophages also decreased after TFPI shRNA treatment (Fig. 7D). However, these effects were reversed by rTFPI treatment (Fig. 7E).

To explore whether TFPI inhibition had therapeutic efficacy against polycythemia, we treated *Jak2V617F*-mutated mice with TFPI mAb (supplemental Fig. 7A). TFPI mAb treatment increased thrombus size and weight but did not influence the survival rate of *Jak2V617F*-mutated mice (supplemental Fig. 7B, C). However, TFPI mAb treatment induced a decrease in PB RBC numbers and Hb and F4/80+CD169+Vcam-1+ macrophage heme content in *Jak2V617F*-mutated mice (supplemental Fig. 7D, E), while normalizing HCT levels (supplemental Fig. 7F). Furthermore, we constructed a mouse model of hypoxia-induced polycythemia by exposing mice to hypoxia for up to 3 weeks (supplemental Fig. 7G). TFPI mAb treatment similarly decreased PB RBC numbers and Hb (supplemental Fig. 7H), and reduced F4/80+CD169+Vcam-1+ macrophage heme content in hypoxia-induced polycythemia mice (supplemental Fig. 7I). This treatment also normalized HCT as expected (supplemental Fig. 7J). The results found that TFPI mAb treatment reduced erythropoiesis in polycythemia.

## Discussion

The erythroblastic island niche consists of central macrophages and surrounding developing erythroid cells in BM[10,12,38]. Central macrophages participate in erythropoiesis by providing growth factors and iron. Here, we identified a soluble protein TFPI from erythroblasts that regulated central macrophage function in erythroblastic island niche. We found that the expression of the rate-limiting enzyme Fech for heme biosynthesis in central macrophages was down-regulated in TFPI

knockout mice. Therefore, we identified a signal protein TFPI from erythroblasts to direct heme biosynthesis in central macrophages for erythropoiesis improvement in turn.

TFPI is an anticoagulant protein that dampens early phases of coagulation. Here, TFPI expression levels were not higher in plasma of *JAK2V617F* positive samples compared to normal cases, consistent with previous work[39]. However, a higher TFPI expression level was found in *JAK2V617F*-mutated erythroblasts. Therefore, erythroblast-derived TFPI may play a local role in erythropoiesis in bone marrow. A higher TFPI expression level was also found in erythroblasts under hypoxia. Cui et al. and Stavik et al. independently discovered that hypoxia down-regulated TFPI expression in breast cancer cells and endothelial cells via HIF-1α and HIF-2α, respectively[40,41]. HIF-1α is highly expressed in breast cancer cells, while HIF-2α is highly expressed in endothelial cells. However, both HIF-1α and HIF-2α exhibit the lowest expression in erythroid lineages among hematopoietic cells. Therefore, it is possible that erythroblasts may employ a unique pathway to increase TFPI expression, which differs from that of breast cancer cells and endothelial cells. Further investigation is required to explore the mechanism underlying hypoxic effect on TFPI expression in erythroblasts. We found that TFPI secreted from erythroblasts improved erythropoiesis through central macrophages. Previous studies have reported the crucial involvement of macrophages in erythropoiesis under both steady-state and stress conditions. Depletion of macrophages, on one hand, leads to impaired physiological erythropoiesis and induced anemia[11,42,43]. On the other hand, macrophage depletion prevents mice recovering from induced anemia and improves the phenotype of polycythemia vera[42]. Here, we found that TFPI had no effect on erythropoiesis after macrophage depletion, suggesting that macrophages are the target cells of TFPI. Therefore, we discover a crucial erythropoiesis regulator TFPI, which mediates the crosstalk between erythroblasts and macrophages.

Heme, which is composed of iron and the organic molecule protoporphyrin, is the essential cofactor of hemoglobin in erythroid cells. Heme biosynthetic changes in erythroid cells result in erythropoietic disorders. Deficiency of UROS, a heme synthetic enzyme, leads to chronic hemolytic anemia in congenital erythropoietic porphyria[44]. The succinyl-CoA deficiency in isocitrate dehydrogenase 1-mutant hematopoietic cells attenuates heme biosynthesis and blocks

erythroid differentiation at the late erythroblast stage[45]. Heme also participates in a variety of physiological roles in macrophages. Heme catabolism in tumor associated macrophages shapes a prometastatic tumor microenvironment to favor immunosuppression, angiogenesis and epithelial-to-mesenchymal transition[46]. Heme oxygenase 1, an enzyme responsible for heme breakdown, is implicated in oxidative stress and inflammatory response in macrophages[47]. Almost all mammalian cell types possess heme biosynthetic pathway except mature erythrocytes[48]. In spite of macrophages with expressing heme biosynthetic enzymes, macrophage heme is not thought to affect erythropoiesis in BM before our work. Here, we found that heme in BM central macrophages contributed to erythropoiesis, suggesting that macrophage heme has paracrine function. Why do BM macrophages provide heme for erythropoiesis besides providing iron? In cardiomyocytes, heme degradation releases free iron to induce cardiac injury[49]. Local iron reduces self-renewal and increases differentiation of hematopoietic stem cells in BM[50]. This phenomenon may represent a protective mechanism by which macrophages transfer heme iron, but not free iron, to erythroblasts in BM. Macrophage depletion in mice with polycythemia vera results in a block in erythroblast differentiation at III and IV stages[42], indicating the transition from III and IV stages to V stage is a key point in erythropoiesis. Heme production in erythroblasts reaches a maximum in the stage III and IV to promote hemoglobin production and erythroblast maturation[51,52]. Here, we found that TFPI deficiency led to the inhibition of heme synthesis in central macrophages. Macrophages-specific knockout of Fech, the terminal heme synthesis enzyme, also prevented hemoglobin synthesis and blocked erythroblast differentiation at III and IV stages. Therefore, the inhibition of heme synthesis in central macrophages of $TFPI^{f/f;EpoR}$ mice results in a block at III and IV stages of erythroblast development. It demonstrates that central macrophage-derived heme also promotes hemoglobin production and erythroblast maturation.

TFPI forms a stable complex with TF/fVIIa to inhibit coagulation[53]. Furthermore, protein S acts as a cofactor for TFPI to accelerate coagulation inhibition[54]. The very low density lipoprotein receptor interacts with TFPI to regulate apoptotic, antiangiogenic, and antitumor activity[55]. Here, we identified a single transmembrane receptor Thbd as a functional receptor of TFPI. Thbd was first determined as a ligand for thrombin and a critical cofactor for the major natural anticoagulant protein C system[56]. Recently, Thbd was also implicated in inflammation, migration, angiogenesis, and leukocyte adhesion[56]. In infiltrating macrophages of aortic aneurysm, Thbd regulates migration, matrix metalloproteinase activities, and oxidative stress[57]. Here, we found that Thbd mediated heme synthesis in macrophages through the aPC/ERK1/2/GATA1 pathway, illustrating an intracellular pathway of Thbd in macrophages.

It has been known for a long time that coagulation system is activated in polycythemia[58]. Thrombosis and major hemorrhage are frequent symptoms of polycythemia[59,60]. This finding suggests that the coagulation system plays an important role in polycythemia. TFPI was first identified as a primary inhibitor of the initiation of blood coagulation and modulates bleeding and clotting[53]. Further investigation found that TFPI is a multivalent protein implicated in bacterial sepsis[61], metastatic tumor growth[55,62], atherosclerosis[63,64], and *Clostridioides difficile* infection[65]. Here, we found that TFPI knockout inhibited erythropoiesis, illustrating a regulator exists between coagulation and erythropoiesis. Furthermore, we illustrated that TFPI knockout increased the percentages of polychromatic and orthochromatic erythroblasts, while decreased the percentages of reticulocytes and RBCs, suggesting that TFPI affected the development of erythroid cells. Two antibodies Concizumab, and Marstacimab that target TFPI have been in clinical evaluation for hemophilia care because of their ability to modulate blood coagulation[66,67]. On the basis of the present results, the inhibition of TFPI would interfere with erythropoiesis. Inhibition of TFPI may be especially suitable to therapy polycythemia with complication of hemorrhage.

In summary, we identified TFPI as a regulator from erythroid cells to increase heme production through binding with its receptor Thbd in central macrophages of BM, which mediated erythropoiesis by providing heme. Our results reveal a signal pathway of the coagulation system that affects erythropoiesis and represents a potential therapeutic strategy for polycythemia.

## Methods

### Ethical statement
Animal experiments were proved by the Zhejiang University Institutional Animal Care and Use Committee. Human study was proved by the Ethics Committee of Second Affiliated Hospital of Zhejiang University. All ethical guidelines were adhered to whilst carrying out this study.

### Human samples
This study included 18 healthy donors (20-68 years old, mean = 40; 11 females and 7 males), and 21 $JAK2^{V617F}$-mutated patients (16–65 years old, mean = 37; 11 females and 10 males) from the Second Affiliated Hospital of Zhejiang University School of Medicine (Table S2). Informed consent was obtained from all subjects. Healthy donors *and $JAK2^{V617F}$*-mutated patients were not economically compensated for the samples donated. The study was approved by the Ethics Committee of Second Affiliated Hospital of Zhejiang University (No. 20230705). Blood samples were collected and the protein levels of TF and TFPI in plasma were measured by ELISA kit according to the manufacturer's instructions (R&D Systems, Minneapolis, MN, USA).

### Animals
C57BL/6 mice were purchased from Zhejiang Provincial Laboratory Animal Center. EpoR-Cre mice were kindly provided by Stuart H. Orkin (Harvard Medical School, Boston, MA)[68]. LysM-Cre (Stock# 004781) and $Jak2^{V617F/+}$ (Stock# 031658) mice were purchased from The Jackson Laboratory. Vav-iCre (Stock# C001019) and $TFPI^{f/f}$ (Stock# S-CKO-06215) mice were purchased from Cyagen Biosciences Inc. (Suzhou, China). CD169-Cre (Stock# NM-KI-215032) and $Thbd^{f/f}$ (Stock# NM-CKO-2101896) mice were obtained from Shanghai Model Organisms Center, Inc. (Shanghai, China). To generate erythroid lineage-specific and macrophage-specific TFPI knockout mice, EpoR-Cre and CD169-Cre mice were crossed with $TFPI^{f/f}$ mice on a C57BL/6 background. LysM-Cre mice were crossed with $Thbd^{f/f}$ mice to generate macrophage-specific Thbd knockout mice. CD169-DTR heterozygous ($CD169^{DTR/+}$) mice on a C57BL/6 background[69], which were generated with DTR complementary DNA (cDNA)[70], were bred in house by crossing $CD169^{DTR/DTR}$ mice with C57BL/6 mice. Vav-iCre mice were crossed with $Jak2^{V617F/+}$ mice to generate $Vav-iCre;Jak2^{V617F/+}$ mice ($Jak2^{V617F}$). $TFPI^{f/f;EpoR}$ mice were crossed with CD169-DTR mice to generate $TFPI^{f/f;EpoR};CD169^{DTR/+}$ mice. All mice were housed under 12h-light-dark cycle with controlled temperature (22 ± 2 °C) and humidity (30-70%) in a specific pathogen-free barrier facility. Experiments were performed on 6-8-week-old mice. Both male and female mice were used in our study. The experimental conditions and procedures were approved by the Zhejiang University Institutional Animal Care and Use Committee and were consistent with the National Institutes of Health Guide for the Care and Use of Laboratory Animals.

Fech gene contains 11 exons, and exon 6 was selected as a conditional knockout region. PCR-generated homology arm and conditional knockout region were used to design targeting vector. Cas9, gRNA and targeting vector were co-injected into zygotes for the generation of $Fech^{f/f}$ mice. Mouse pups were genotyped by PCR and verified by sequencing. CD169-Cre mice were crossed with $Fech^{f/f}$ mice to generate macrophage-specific Fech knockout mice.

## Reagents

Recombinant mouse TFPI proteins (mouse rTFPI; R&D Systems) were injected intravenously (i.v.) into mice at a dose of 50 µg/kg for 5 days. mouse rTFPI and human rTFPI (R&D Systems) were added to the cell culture medium at a concentration of 200 ng/ml. Anti-TFPI monoclonal antibodies (Concizumab, Creative Biolabs, Shirley, New York, NY, USA) were injected i.v. into mice at a dose of 5 mg/kg. Anti-TF monoclonal antibodies were generated and injected i.v. into mice at a dose of 20 mg/kg. SCH772984, a ERK1/2 inhibitor, was injected intraperitoneally (i.p.) into mice at a dose of 10 mg/kg for 7 days. AS1842856, a FOXO1 inhibitor, was injected i.p. into mice at a dose of 10 mg/kg for 7 days. JNK-IN-8, a JNK inhibitor, was injected i.p. into mice at a dose of 10 mg/kg for 7 days. In BrdU incorporation assays, 1 mg of BrdU was administered to mice by i.p. injection, $Ter119^+CD71^+$ cells were collected and processed according to the manufacturer's instructions in the BrdU Kit (BD Biosciences, San Jose, CA, USA).

## Complete blood count analysis

Mice were bled to collect ~25 µL via the tail vein to collect blood in EDTA-coated BD Microtainer Blood Collection Tubes (BD, San Jose, CA, USA). Blood was diluted 1:20 in PBS and complete blood counts were measured on an Automatic Blood Analyzer (Sysmex, Kobe, Japan).

## TF procoagulant activity

Mouse plasma was diluted with HBSA buffer (137 mm NaCl, 5.38 mM KCl, 5.55 mM glucose, 10 mM HEPES, 0.1% bovine serum albumin, pH 7.5), and microparticles (MPs) were pelleted at 20,000 g for 30 min. Resuspended MPs were incubated with mouse TF monoclonal antibody. Then, 50 µL HBSA containing mouse FVIIa and human plasma−derived factor X was added incubated for 2 h at 37 °C. The reactions were quenched in EDTA buffer and chromogenic substrate Pefachrome FXa 8595 was added. Finally, absorbance at 405 nm was measured to calculate procoagulant activity (PCA).

## Partial ligation of the inferior vena cava (IVC)

Mice were anesthetized and IVC was exposed via sterile laparotomy. After soaking intestines in warm saline, all the IVC side branches were ligated, and then the IVC was gently isolated from aorta. A suture was placed over the IVC, and ligated over a 30-gauge blunt needle, and then the needle was removed. Finally, the abdominal wall closed with a simple continuous suture. thrombi were excised and weighted by scale after 4 h.

## Real-time quantitative polymerase chain reaction (RT-qPCR)

Total RNA from tissue and cell samples were isolated using RNAiso reagent (TaKaRa, Dalian, China). After treatment with DNase I (Roche, Basel, Switzerland), reverse transcription was performed using AMV reverse transcriptase (TaKaRa) to obtain cDNA. Primers are listed in the Table S3. 18 S rRNA was used as an internal reference gene. RT-qPCR was performed using an ABI StepOne Real-Time PCR System (Applied Biosystems, Foster City, CA) with TB Green Premix Ex Taq II (TaKaRa).

## Western blot and co-immunoprecipitation (Co-IP) assays

For Western blot, tissue and cell samples were homogenized in lysis buffer (20 mM HEPES, 1.5 mM $MgCl_2$, 0.2 mM EDTA, 100 mM NaCl, 0.2 mM dithiothreitol, 0.5 mM sodium orthovanadate, 0.4 mM PMSF, pH 7.4) containing phosphatase inhibitor (phosphatase inhibitor cocktail; Sigma-Aldrich). The soluble protein concentration was measured using the Bradford method. Proteins (20 µg of each sample) were separated by SDS-PAGE and electroporated onto polyvinylidene difluoride (PVDF) membranes (Fig. 6G-J). Then, non-fat milk was blocked, primary and secondary antibodies were incubated, and ECL

reactions were performed. Band intensities on blots were quantified using ImageJ software.

For Co-IP, cells were washed with cold PBS and then lysed using lysis buffer for 1 h. The supernatants were collected after centrifugation 14,000 x g for 10 min at 4 °C and incubated with antibody-coupled Protein A beads or Protein G beads (Sigma-Aldrich) according to the manufacturer's instructions. The beads were washed three times with lysis buffer, followed by Western blot analysis.

The following primary antibodies were used for Western blot. Anti-mouse TFPI antibody (1:500, ab180619, Abcam, Cambridge, UK), anti-human TFPI antibody (1:500, ab260042, Abcam) anti-mouse Fech antibody (1:1000, 14466-1-AP, Proteintech, Wuhan, China), anti-mouse Thbd antibody (1:1000, ab230010, Abcam), anti-human Thbd antibody (1:1000, ab109189, Abcam), anti-mouse GATA1 antibody (1:1000, sc-265, Santa Cruz Biotechnology, Santa Cruz, CA, USA), anti-mouse PC antibody (1:1000, ab313386, Abcam), anti-Flag antibody (1:1000, F3165, Sigma-Aldrich), anti-Myc antibody (1:1000, 2276, Cell Signaling Technology), anti-GST antibody (1:1000, 2622, Cell Signaling Technology), anti-p-GATA1 antibody (1:500, PA5-104243, Thermo Fisher Scientific, Waltham, MA, USA), anti-β-actin antibody (1:1000, sc-47778, Santa Cruz Biotechnology) and anti-GAPDH antibody (1:1000, sc-365062, Santa Cruz Biotechnology). The following secondary antibodies were used for Western blot. Goat anti-rabbit IgG H&L (HRP) (1:2000, ab6721, Abcam), Goat Anti-mouse IgG H&L (HRP) (1:2000, ab205719, Abcam).

## GST pull-down assay

For tagging sequences with Flag, the cDNA encoding the sequence of TFPI was cloned into the pcDNA3.1 vector and then transfected into HEK293T cells in a 100-mm culture dish. GST-tagged Thbd protein was expressed in *E. coli*. BL21 (ATCC BAA-1025), the protein was purified and the GST-tagged Thbd protein was incubated with GSH-agarose in binding buffer (50 mM Tris/HCl, 150 mM NaCl, 1 mM EDTA, 0.5% NP40, 10% glycerol, pH 7.4), and rotated at 4 °C for 1 h. Then, the beads loaded with the GST-tagged Thbd protein were collected and incubated with the Flag-tagged TFPI protein at 4 °C for 2 h. The beads were washed 3 times followed by Western blot.

## PHZ treatment

For induction of hemolytic anemia, mice were injected i.p. with PHZ (Sigma-Aldrich) at a dose of 40 mg/kg on days 0 and 1 of the experiment. Peripheral blood was collected 4 days before the start of treatment and on days 4, 7 and 12 after treatment.

## Hypoxia exposure

Mice were exposed to hypoxia simulating an altitude of 5000 m (54.02 kPa, 10.8% $O_2$) in a well-ventilated hypobaric chamber (Guizhou Fenglei Air Ordnance Co., Ltd.). Control mice were set at sea level (100.08 kPa, 20.9% $O_2$) in the same chamber.

## Flow cytometry and cell isolation

BM cells were isolated by thoroughly flushing tibias, femurs, and humeri using a 5 ml polystyrene tube with a strainer (BD Biosciences). Spleens were mashed through a 70 µm nylon filter. Isolation of in vivo-formed EBIs was described previously[11,71,72]. Briefly, BM was flushed gently with Iscove's modified Dulbecco's medium (IMDM) containing 3.5% sodium citrate and 20% fetal calf serum solution using an 18 G syringe without centrifugation. Spleen was cut into small pieces, and incubated in RPMI1640 containing 0.075% Collagenase IV and 0.004% DNase I for 30 min. The suspension was passed through an 18 G syringe, washed and resuspended in IMDM containing 3.5% sodium citrate and 20% fetal calf serum solution. Cells were labeled with fluorochrome-conjugated antibodies in staining buffer for 30 min at 4 °C. Samples were analyzed on a Gallios flow cytometer (Beckman

Coulter, Miami, FL, USA). The analysis was performed using FlowJo software (Tree Star, Ashland, OR, USA).

Erythroid lineages were labeled with antibodies directed at CD71, Ter119 and CD44. F4/80⁺CD169⁺Vcam-1⁺ macrophages were labeled with antibodies directed at Ter119, Ly6G, F4/80, Vcam-1, and CD169. EBIs were labeled with antibodies directed at F4/80⁺Ter119⁺ multiplet population.

The following fluorescently labeled antibodies (BioLegend, San Diego, USA) were used: PE-anti-TER-119/Erythroid cells (clone Ter-119, 1:100), PE/Cyanine7-anti-CD71 (clone RI7217, 1:100), APC-anti-CD44 (clone IM7, 1:100), PE-anti-Ly6G (clone 1A8, 1:100), FITC-anti-TER-119/Erythroid cells (clone Ter-119, 1:100), BV421-anti-F4/80 (clone BM8, 1:25), APC-anti-c-Kit (clone 2B8, 1:50), anti-mouse lineage cocktail (FITC-anti-CD4 [clone GK1.5, 1:200], FITC-anti-NK1.1 [clone PK136, 1:100], FITC-anti-CD11b [clone M1/70, 1:200], FITC-anti-B220 [clone-neRA3-6B2, 1:100], FITC-anti-Gr-1 [clone RB6-8C5, 1:200], FITC-anti-CD8a [clone 53-6.7, 1:100], FITC-anti-TER-119/Erythroid cells, [clone Ter-119, 1:100]), and APC-anti-Vcam-1 (clone 429, 1:100). Flow analysis of live cells by exclusion of dead cells using propidium iodide (PI, Sigma-Aldrich). Identification of apoptotic cells were carried out using the FITC Annexin V Apoptosis Detection kit (BioLegend). For sorting of Lin⁻c-kit⁺CD71⁺ cells, Ter119⁺CD71⁺ cells, Ter119⁺ cells, erythroblasts and F4/80⁺CD169⁺Vcam-1⁺ macrophages, samples were processed under sterile conditions and sorted on Fluorescence-activated cell sorting (FACS) sorting with Moflo Astrios EQ (Beckman Coulter).

## Colony-forming unit (CFU) assay

For CFU-E assay, the Lin⁻c-kit⁺CD71⁺ cells were flow sorted and plated in erythropoietin-containing methylcellulose culture medium (StemCell Technologies, Vancouver, BC, Canada) and incubated at 37 °C in 5% $CO_2$ humidified atmosphere for 7 days. The number of colonies formed on each plate was counted using an inverted microscope.

## Measurement of heme and Hb content

Intracellular heme content was determined according to fluorometric assays. Cells were harvested and resuspended in 2 M oxalic acid and heated at 100 °C for 30 min to remove iron from heme. The resultant protoporphyrin was measured by fluorescence (400 nm excitation and 662 nm emission). Endogenous protoporphyrin content was measured by detecting fluorescence in oxalic acid-treated unheated cells. The Hb content was quantified with the Drabkin's reagent (Sigma-Aldrich).

## Cell culture and transfection

HEK293T cells were obtained from American Type Culture Collection (ATCC, #CRL-3216) and were cultured in media composed of Dulbecco's Modified Eagle's Medium (DMEM), 10% fetal bovine serum (FBS) and 1% penicillin/streptomycin. Ter119⁺CD71⁺ cells were sorted and cultured in media composed of IMDM, 10% FBS, 1% bovine serum albumin, 0.2 mg/mL holotransferrin, 10 mg/mL insulin and erythropoietin at the different concentrations. F4/80⁺CD169⁺Vcam-1⁺ macrophages were sorted and cultured in media composed of RPMI 1640, 10% FBS, 10 mM HEPES, and 10 ng/mL macrophage colony-stimulating factor (G-CSF). Cell viability was determined using trypan blue and counted with a hemocytometer. Cells were incubated at 37 °C in 5% $CO_2$ humidified atmosphere. HEK293T cells and F4/80⁺CD169⁺Vcam-1⁺ macrophages were transfected with pcDNA3.1 vector carrying the cDNA encoding sequence of TFPI, Thbd, or GATA1.

## Macrophage depletion

To deplete CD169⁺ macrophages, heterozygous *CD169^{DTR/+}* mice were injected i.p. with diphtheria toxin (Sigma-Aldrich) at a dose of 10 µg/kg. In some experiments, macrophages were depleted by intravenous injection 200 µl of clodronate liposomes three times a week. For PHZ experiments, mice were injected with clodronate liposomes on day -2 and day 0.

## Lentivirus production and infection

For generation of mouse TFPI, aPC, GATA1, Thbd, and human TFPI lentiviral vectors for knockdown, shRNA sequences targeting specific genes were synthesized and cloned into pLKO.1 vectors. The vectors were co-transfected into HEK293T cells with pSPAX2 (Addgene, Cambridge, MA, USA) and pMD2.G (Addgene, Cambridge, MA, USA) for packaging of lentiviral vectors. Lentiviral supernatants were collected 48 h post-transfection. For Ter119⁺CD71⁺ cell and F4/80⁺CD169⁺Vcam-1⁺ macrophage infection, cells were transduced with lentivirus at a multiplicity of infection (MOI) of 10 and selected with puromycin (8 µg/ml) for 48 h. The knockdown efficiency was assessed by RT-qPCR and Western blot. Mice were injected with lentivirus at the dose of $6 \times 10^8$ pfu by tail vein injection. Target sequences are listed in Table S4.

## Enzyme-linked immunosorbent assay (ELISA)

Femoral BM was rinsed with PBS and centrifuged to obtain cell supernatant. Blood samples were treated with sodium citrate and centrifuged at 4 °C to extract plasma. The protein expression of TF and TFPI in plasma were measured by TF and TFPI ELISA kit according to the manufacturer's instructions (R&D Systems). The protein expression of TFPI-2 in BM supernatant or plasma were measured by TFPI-2 ELISA kit according to the manufacturer's instructions (USCN Life Science Inc., Wuhan, China). Plasma TAT complexes were measured by thrombin-antithrombin complexes ELISA kit (Abcam, Cambridge, UK).

## In vitro human EBI formation

CD34⁺ cells were sorted by CD34 MicroBeads (Miltenyi Biotec, Germany) from human cord blood. Macrophages were derived from CD34⁺ cells by culturing in IMDM medium containing 2% human peripheral blood plasma, 3% human AB serum, 3 IU/mL heparin, 10 µg/mL insulin, 10 ng/mL stem cell factor (SCF), 1 ng/mL interleukin-3 (IL-3), 100 ng/mL macrophage colony-stimulating factor (M-CSF), 50 ng/mL fms-like tyrosine kinase 3 (FLT3), and 1 × penicillin-streptomycin. Erythroblasts were also derived from CD34⁺ cells. The cell culture procedure was comprised of 3 phases and 2 phased were used in present. In day 0 to day 6, CD34⁺ cells were cultured in IMDM containing 2% human peripheral blood plasma, 3% human AB serum, 200 µg/mL holo-human transferrin, 3 IU/mL heparin, 10 µg/mL insulin, 10 ng/mL SCF, 1 ng/mL IL-3, and 3 IU/mL erythropoietin for 6 days. In day 7 to day 11, IL-3 was omitted from the culture medium. The Day 11 erythroblasts were pretreated with TFPI shRNA or control shRNA and mixed with macrophages at a 20:1 ratio. Then cells were cultured for 12 h in IMDM containing 2% human peripheral blood plasma, 3% human AB serum, 3 IU/mL heparin, 10 µg/mL insulin, 200 µg/mL holo-human transferrin, 10 IU/ml EPO, 5 mM $Mg^{2+}$, and 5 mM $Ca^{2+}$. $1 \times 10^5$ cells were collected for cytospin analysis.

## Transduction of JAK2^{V617F} mutation into CD34⁺ cells

The JAK2-transduced CD34⁺ cells were prepared as previously described[73]. Briefly, human wild type or mutant JAK2 cDNAs were respectively cloned into the MIGR1-IRES-GFP vector (Addgene, Cambridge, MA, USA). The vectors were co-transfected with lentivirus packaging plasmids pMD.G into HEK293T cells with Lipofectamine 3000. After 48 h, the lentiviral supernatants were collected, concentrated and stored at −80 °C. For infections, CD34⁺ cells were incubated with 50 µl of viral stock for 48 h.

## RNA sequencing analysis

RNA was extracted from sorted F4/80⁺CD169⁺Vcam-1⁺ macrophages of *TFPI^{f/f}* and *TFPI^{f/f;EpoR}* mice, or BM cells of normoxia- and hypoxia-exposed mice. RNA quality was assessed by an Agilent 2100 Bioanalyzer (Agilent, Palo Alto, CA, USA) and quantified by a Nanodrop ND-2000 Spectrophotometer (Thermo Scientific, Waltham, MN, USA) prior to sequencing. High-quality RNA samples were used to construct

sequencing libraries. RNA-seq transcriptome libraries were prepared using 1 μg of total RNA using the TruSeq RNA Sample Prep Kit from Illumina (San Diego, CA, USA). Libraries were sequenced using Illumina Novaseq 6000 with 2 × 151 bp read length. Expression levels for each transcript were using the fragments per kilobase of exon per million mapped reads (FPKM) method. Secreted proteins were identified as proteins carrying a signal peptide but lacking a transmembrane region[74]. Gene Oncology (GO) enrichment analysis were performed for the differentially expressed genes (DEGs) in the DAVID resource (https://david.ncifcrf.gov/). Kyoto Encyclopedia of genes and genomes (KEGG) path analysis of DEGs were carried out through clusterprofiler package in R. The ggplot2 package in R were used for Heatmap generation.

### Quantification and statistical analysis
Data were shown as mean ± standard error of the mean (SEM). The biological repeats were indicated by 'n'. Statistical analysis was performed using one-way ANOVA with SPSS (version 23) software. When the variances were significantly different ($P < 0.05$), logarithmic transformation was used to stabilize the variance. If the data did not have a normal distribution, statistical significance was evaluated using the Mann-Whitney U-test (two-tailed). $P$ values < 0.05 was considered statistically significant.

### Reporting summary
Further information on research design is available in the Nature Portfolio Reporting Summary linked to this article.

## Data availability
The RNA-seq data under this study are available at GEO under accession code GSE224993. and GSE224994. The data generated in this study are provided in the Source Data file. A Figshare archive is present with this paper https://doi.org/10.6084/m9.figshare.23715525. Source data are provided with this paper.

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

## Acknowledgements

We thank Stuart H. Orkin and Yuko Fujiwara for providing EpoR-Cre mice. This work was supported by the Program for the Natural Science Foundation of China (41776151, 81971874), the Ten thousand plan youth talent support program of Zhejiang Province, the Zhejiang Provincial Natural Science Foundation of China (LZ23C110001).

## Author contributions

Q.Z., T.L., and X.J.L. conceived and supervised the study. J.K.M., L.D.S., and J.R.H. collected clinical samples and data. J.K.M, L.D.S., and L.L.F. performed most of the experiments. J.L.L., L.P., Q.D., Y.L., and X.Q.C. helped with experiments. J.K.M, L.L.F., T.S., X.L.Z., S.Y.C., S.Y., Q.Z., T.L., and X.J.L. performed the data analysis. J.K.M., Q.Z., T.L., and X.J.L. wrote the manuscript.

## Competing interests

The authors declare no competing interests.
