## [Peer Review File · Nature Communications]

TFPI from Erythroblasts Drives Heme Production in Central Macrophages Promoting Erythropoiesis in PolycythemiaREVIEWER COMMENTS

Reviewer #1 (Remarks to the Author):

In this manuscript 'TFPI from Erythroblasts Drives Heme Production in Central Macrophages Promoting Erythropoiesis in Polycythemia', the authors tried to use the JAK2V617F mutation, hypoxia and hemolytic anemia models to study polycythemia, together with different type of gene knockout mice.

In general, the authors fail to explain their scientific study clearly. The organization of results is quite confusing and some of the results are lack of key information. The language also needs attention.

Some major comments (not a full list)

1. Please, define the term "central macrophages" in the main text, are they macrophages in the bone marrow and what differentiate them from other macrophages? After macrophage depletion with diphtheria toxin or clodronate liposomes, the authors should show that the macrophages in the bone marrow are depleted by immunohistochemistry, immunofluorescence or flow.
2. In result 2, the authors isolated mouse erythroid cells to knock down TFPI in vitro, first of all they should show which type of mice were used in this in vitro study. And TFPI protein should be measured before and after knockdown to show that the negative result is because of failed knockdown or lack of macrophages as they claimed. Further, it is better to show the crosstalk between macrophages and erythroid cells in this in vitro study by co-culturing the two cell types in vitro.
3. It has been shown that hypoxia downregulates TFPI (Cui et al. 2016 and Stavik et al. 2016), please address this controversy in the discussion part. The authors should explain why TFPI was measured in all bone marrow cells, not erythroid cells, which is important for their scientific question. And since mice were exposed to 10% O₂ for 1 week, RBC, Hb, HCT and erythroid colony should be measured to show the effect of hypoxic condition on erythroid generation. In the methods, the information of hypoxic chamber used in the mice study should be shown. All together, these hypoxia experiments seems not relevant to the main scientific question- TFPI regulate erythroid generation through macrophage in polycythemia, so should be left out.
4. Describe the procedure of TFPI knockdown in CD169^{DTR/+} mice. Is it erythroid-specific knock down? If it is not, how is TFPI protein level in plasma after knockdown? Did the authors use lentivirus? Lentivirus is good for cells, not for mice, so how is the transfection efficiency?
5. In result 2 and figure 2, explain the abbreviation DT, is it diphtheria toxin? The authors used two different methods to deplete macrophages in two mice models, explain why you cannot use the same to keep the experiments consistent and more comparable.
6. In figure 2F, it seems the macrophage depletion itself affected erythroid generation, the authors should analyze and explain it. Is this effect disappeared when TFPI is absent? The authors should do more work on this important issue.
7. In figure 2G, in the PBS control group, the percentage of erythroblast significantly increased in TFPI^{f/f} EpoR mice in stage III and IV, but dropped down in stage V, explain why.
8. In Figure 3J, the authors claimed that 'We found that the phosphorylation level of GATA1 in Ter119+ 207 cells and central macrophages increased over time', but they only showed central macrophages and didn't mention which mice model the macrophages were isolated from.
9. In the results 'TFPI interacts with Thbd in central macrophages', the authors showed TFPI interacted with Thbd in BM cells by co-IP, but in order to confirm their hypothesis, central macrophages should be

used instead.

Reviewer #2 (Remarks to the Author):

In this paper, Ma et al. found that TFPI, an anti-coagulant protein, regulates erythropoiesis via controlling heme biosynthesis in erythroblastic island (EBI) central macrophage. They used TFPI tissue-specific knock-out (KO) mice and polycythemia mouse models. The functions of TFPI were linked with EBI central macrophage. The authors found that TFPI on the erythroblast surface interacted with Thbd expressed on the central macrophage surface, and the interaction activated the p-ERK/GATA1/Fech pathway to control heme biosynthesis in the central macrophage. These findings were further proved with several mouse modes, including Thbd and Fech tissue-specific KO mice. Finally, they explore the applications of targeting TFPI to treat polycythemia. They concluded that TFPI from erythroblasts drives heme biosynthesis in central macrophage to support erythropoiesis in polycythemia and can be targeted for treating polycythemia.

While the manuscript presented numerous datasets using different model systems, the underlying logic is unclear. It is important to resolve the following concerns for the story to be published.

1, Thrombosis in polycythemia is complex and involves platelets, red blood cells, and many other factors. TFPI is an inhibitor of coagulation. TFPI antibodies in clinical trials were evaluated based on TFPI's roles in inhibiting coagulation. From this perspective, inhibiting TFPI in polycythemia may further promote thrombosis and worsen the disease. In this study, what about the survival of those mice and their thrombosis treated with TFPI antibody in supplemental Figure 7? These data should be shown. The same questions apply to the JAK2V617F-TFPI f/f VaV mice in Figure 1K-1L.

2, Even though the data demonstrates that TFPI plays roles in erythropoiesis via TFPI-Thbd-mediated erythroblast-macrophage interaction, it does not exclude TFPI's intrinsic roles in erythropoiesis. TFPI's intrinsic roles in erythroblast should be included in the manuscript, and they can be monitored after knock-out or over-expression of TFPI in erythroblast without macrophage co-culture.

3, What about the EBI number changes in bone marrow and spleen of those mice, including TFPI KO mice, Thbd KO mice, and rTFPI treated mice? These results should be included to make their conclusions stronger.

4, Many shRNAs were used in the study. ShRNAs commonly have off-target effects, CRISPER-CAS9 system is preferred. Even with the shRNAs, at least the western blot showing the efficiency of those shRNAs should be included.

5, in Figure 1A, TFPI expression levels were not higher in JAK2V617F positive samples compared to normal cases, which needs to be explained.

6, in Figure 1D, the difference in TFPI expression levels seems related to JAK2V617F mutation, other than co-culture of macrophage and erythroblast.

7, in Figure 1I, apoptotic cell percentages in the control group are more than 60%. The data is not reliable and should be repeated. The same problems are in supplemental Figure 3E, supplemental Figure 4I, and Figure 5F.

8, in Figure 2A-2B, TFPI KO cells, instead of shRNA-mediated knock-down, are better to be used since TFPI tissue-specific KO mice are available, and the KO cells can be purified directly from mice.

9, In Figure 2D, the picture on the right is not consistent with the three left statistic figures since it

looks like there is a significant difference between the third and fourth tubes.

10, Was TF expressed on central macrophage? In Figure 4A, TF expression levels should be tested on central macrophage instead of total bone marrow. Similarly, in Figure 6A, aPC mRNA levels should be tested in different lineage cells instead of total bone marrow since total bone marrow is a mixture of cells.

11, in Figure 6E, the significance of comparing the second and fourth groups should be shown in the figure to support their conclusion "SCH772984 treatment resulted in no further reduction in heme content after GATA1 shRNA treatment (Figure 6E)".

12, in Figure 6G and 6H, the blot of p-GATA1 should be repeated since the current data do not support their conclusion "We then found that the protein level of p-GATA1 was decreased in Thbd f/f;LysM or SCH772984-treated mice".

13, in Figure 6L, it seems that the fourth group is higher than the second group, which do not support their conclusion "this function was abrogated by SCH772984 and GATA1 shRNA treatment. (Figure 6J-6L)". The significance of comparing the second and fourth groups should be shown in the figure.

14, The working model in Figure 6M showed that GATA1 binds the gene and promotes Fech's expression. It seems that GATA1 is a direct transcription factor of Fech. The original study should be cited if this finding was published before, or data should be shown to prove it if it was found by the authors.

15, Currently, the identification of EBI macrophage is not fully understood. The marker combination Ly6G/Ter119+F4/80+CD169+Vcam-1+ used in this manuscript cannot represent all EBI macrophages, so the statement of "central macrophage" to refer to the sorted macrophages is not accurate.

A point-by-point response to reviewers

We thank the editor and reviewers for their valuable and helpful comments. To address their questions, we performed additional experiments and made changes in the manuscript. The corresponding changes in the manuscript are highlighted in blue.

Reviewer #1 (Remarks to the Author):

In this manuscript 'TFPI from Erythroblasts Drives Heme Production in Central Macrophages Promoting Erythropoiesis in Polycythemia', the authors tried to use the JAK2V617F mutation, hypoxia and hemolytic anemia models to study polycythemia, together with different type of gene knockout mice.

In general, the authors fail to explain their scientific study clearly. The organization of results is quite confusing and some of the results are lack of key information. The language also needs attention.

Response: We thank the reviewer's helpful and constructive suggestions for improvement. Within the scope of this revision, we refined the presentation of our results. We have thoroughly examined each of the points raised and added relevant data and information to provide a comprehensive response. Besides, we carefully optimized the language used in the article. The corrections to the typical language issues mentioned by reviewers are shown in the table below.

Line	correction
165	"sorted from TFPI^{ff} mice" was added.
175	"diphtheria toxin (DT) " was added.
242-243	"We found that the phosphorylation level of GATA1 in Ter119 ⁺ cells and central macrophages increased over time" was corrected to "We found that the phosphorylation level of GATA1 in F4/80 ⁺ CD169 ⁺ Vcam-1 ⁺ macrophages of TFPI^{ff};EpoR mice increased over time after rTFPI treatment".
350-351	"this function was abrogated by SCH772984 and GATA1 shRNA treatment" was corrected to "this effect was attenuated by SCH772984 and GATA1 shRNA treatment".
122,174, 201, etc.	"central macrophage" was corrected to "F4/80 ⁺ CD169 ⁺ Vcam-1 ⁺ macrophage".

The point-by-point responses to the reviewer's comments and questions are provided below.

Some major comments (not a full list)

1. Please, define the term "central macrophages" in the main text, are they macrophages in the bone marrow and what differentiate them from other macrophages? After macrophage depletion with diphtheria toxin or clodronate liposomes, the authors should show that the macrophages in the bone marrow are depleted by immunohistochemistry, immunofluorescence or flow.

Response: We thank the reviewer for this comment which helps us to improve our study.

Central macrophages act as nurse cells for erythroblasts and are integral components of erythroblastic islands (EBIs), which were firstly identified in bone marrow^{10,11,12,13}.

Later investigation identified EBIs in the murine spleen and fetal liver, highlighting the essential nature of central macrophages for promoting erythropoiesis¹⁴. Unlike other macrophages, central macrophages promote erythropoiesis by providing nutrients and signals to surrounding erythroblasts, as well as phagocytosing nuclei extruded from

developing erythrocytes^{15,16}. The interaction between central macrophages and erythroblasts is mediated by adhesion molecules, such as Vcam-1 and CD169 on macrophages¹⁷. The introduction was added in Line 70-78.

The flow cytometry results were added to illustrate macrophage depletion in bone marrow in Figure 2D and 2I, Line 174-175 and 180 as result.

Figure 2D. (D) FACS plots of F4/80⁺CD169⁺Vcam-1⁺ macrophages of *CD169^{DTR/+}* mice after DT treatment.

Figure 2I. (I) FACS plots of F4/80⁺CD169⁺Vcam-1⁺ macrophages of *TFPI^{f/f};EpoR* mice after clodronate liposomes treatment.

References:

10. Bessis M. Erythroblastic island, functional unity of bone marrow. *Rev Hematol* **13**,

8-11 (1958).

11. Chow A, *et al.* CD169⁺ macrophages provide a niche promoting erythropoiesis under homeostasis and stress. *Nat Med* **19**, 429-436 (2013).
12. Li W, *et al.* Identification and transcriptome analysis of erythroblastic island macrophages. *Blood* **134**, 480-491 (2019).
13. Romano L, *et al.* Erythroblastic islands foster granulopoiesis in parallel to terminal erythropoiesis. *Blood* **140**, 1621-1634 (2022).
14. Manwani D, Bieker JJ. The erythroblastic island. *Curr Top Dev Biol* **82**, 23-53 (2008).
15. Chasis JA, Mohandas N. Erythroblastic islands: niches for erythropoiesis. *Blood* **112**, 470-478 (2008).
16. Muckenthaler MU, Rivella S, Hentze MW, Galy B. A red carpet for iron metabolism. *Cell* **168**, 344-361 (2017).
17. Li W, Guo R, Song Y, Jiang Z. Erythroblastic island macrophages shape normal erythropoiesis and drive associated disorders in erythroid hematopoietic diseases. *Front Cell Dev Biol* **8**, 613885 (2021).

2. In result 2, the authors isolated mouse erythroid cells to knock down TFPI in vitro, first of all they should show which type of mice were used in this in vitro study. And TFPI protein should be measured before and after knockdown to show that the negative result is because of failed knockdown or lack of macrophages as they claimed. Further, it is better to show the crosstalk between macrophages and erythroid cells in this in vitro

study by co-culturing the two cell types *in vitro*.

Response: We thank the reviewer for this comment which helps us to improve our study.

TFPI^{fl/fl} mice was used in this *in vitro* study, and this information was added in Line 165 and in supplemental Figure 2A. TFPI expression data was also added using Western blot in *Ter119⁺CD71⁺* cells before and after TFPI shRNA treatment in supplemental Figure 2B.

Additionally, we conducted TFPI knockdown in *Ter119⁺CD71⁺* cells isolated from *TFPI^{fl/fl}* mice and co-cultured with central macrophages. The data was added in supplemental Figure 2D and 2E, Line169-172 as results.

supplemental Figure 2A-2E. (A) Schematic diagram of *in vitro* cultivation and proliferation of *Ter119⁺CD71⁺* cells. (B) TFPI protein expression of *Ter119⁺CD71⁺* cells after TFPI shRNA treatment. (C) The expansion of *Ter119⁺CD71⁺* cells after TFPI shRNA treatment. (D) Schematic diagram of proliferation of *Ter119⁺CD71⁺* cells during co-culture with *F4/80⁺CD169⁺Vcam-1⁺* macrophages *in vitro*. (E) The expansion of *Ter119⁺CD71⁺* cells treated with TFPI shRNA during co-culture with *F4/80⁺CD169⁺Vcam-1⁺* macrophages.

3. It has been shown that hypoxia downregulates TFPI (Cui et al. 2016 and Stavik et al. 2016), please address this controversy in the discussion part. The authors should explain why TFPI was measured in all bone marrow cells, not erythroid cells, which is important for their scientific question.

And since mice were exposed to 10% O₂ for 1 week, RBC, Hb, HCT and erythroid colony should be measured to show the effect of hypoxic condition on erythroid generation. In the methods, the information of hypoxic chamber used in the mice study should be shown. All together, these hypoxia experiments seems not relevant to the main scientific question- TFPI regulate erythroid generation through macrophage in polycythemia, so should be left out.

Response: We thank the reviewer for this comment which helps us to improve our study. Cui et al. and Stavik et al. independently discovered that hypoxia downregulated TFPI expression in breast cancer cells and endothelial cells via HIF-1 α and HIF-2 α , respectively^{40,41}. HIF-1 α is highly expressed in breast cancer cells, while HIF-2 α is highly expressed in endothelial cells. However, both HIF-1 α and HIF-2 α exhibit the lowest expression in erythroid lineages among hematopoietic cells (supplemental Figure 1K and 1L). Therefore, it is possible that erythroblasts may employ a unique pathway to increase TFPI expression, which differs from those of breast cancer cells and endothelial cells. Further investigation is required to explore the mechanism underlying hypoxic effect on TFPI expression in erythroblasts. The data was added in supplemental Figure 1K and 1L, the result description was added in Line 150-153, and

discussion in Line 395-404.

We agree with the reviewer's comment that TFPI should be measured in erythroid cells, but not in all bone marrow cells. The result of TFPI expression in erythroid lineages was added in supplemental Figure 1B and 1J, and the result description was added in Line 117-119 and 149-150.

The data of RBC, Hb, HCT and erythroid colony in hypoxia-exposed mice was added in supplemental Figure 1M and 1N and result description in Line 153-154. The information of hypoxic chamber was added in the method and in Line 595. Moreover, the hypoxia experiments in Figure 1 were deleted to make the research more focused.

supplemental Figure 1K and 1L. (K) HIF-1 α mRNA levels of different cell lineages in BM based on Haemosphere (<https://www.haemosphere.org>). (L) HIF-2 α mRNA levels of different cell lineages in BM based on Haemosphere.

supplemental Figure 1B and 1J. (B) TFPI expression in Ter119⁺ cells of *Jak2*^{V617F}-mutated mice. (J) TFPI expression in Ter119⁺ cells of hypoxia-exposed mice.

supplemental Figure 1M and 1N. (M) PB RBC numbers, Hb, and HCT in hypoxia-exposed mice. (N) CFU-E number in hypoxia-exposed mice.

References:

40. Cui XY, *et al.* Effect of hypoxia on tissue factor pathway inhibitor expression in breast cancer. *J Thromb Haemost* **14**, 387-396 (2016).
41. Stavik B, *et al.* EPAS1/HIF-2 alpha-mediated downregulation of tissue factor pathway inhibitor leads to a pro-thrombotic potential in endothelial cells. *Biochim Biophys Acta* **1862**, 670-678 (2016).

4. Describe the procedure of TFPI knockdown in CD169^{DTR/+} mice. Is it erythroid-specific knock down? If it is not, how is TFPI protein level in plasma after knockdown? Did the authors use lentivirus? Lentivirus is good for cells, not for mice, so how is the transfection efficiency?

Response: We thank the reviewer for this comment which helps us to improve our study. In the firstly submitted manuscript, lentivirus was employed for TFPI knockdown. We employed a more targeted strategy by utilizing *TFPI^{f/f;EpoR};CD169^{DTR/+}* mice to study the effect of erythroid-specific TFPI knockout on erythropoiesis under macrophage depletion. The TFPI protein level in plasma was also measured after TFPI erythroid-

specific knockout and the data was added in Figure 2C-2G, and the data description was added in Line 172-179.

Figure 2C-2G. (C) Schematic diagram of double mutant $TFPI^{f/f;EpoR};CD169^{DTR/+}$ mice. (D) FACS plots of $F4/80^+CD169^+Vcam-1^+$ macrophages of $CD169^{DTR/+}$ mice after DT treatment. (E) The plasma TFPI concentration of $TFPI^{f/f;EpoR};CD169^{DTR/+}$ mice. (F) PB RBC numbers, Hb, and HCT in $TFPI^{f/f;EpoR};CD169^{DTR/+}$ mice after DT treatment. (G) BM cell pellets from $TFPI^{f/f;EpoR};CD169^{DTR/+}$ mice after DT treatment.

5. In result 2 and figure 2, explain the abbreviation DT, is it diphtheria toxin? The authors used two different methods to deplete macrophages in two mice models, explain why you cannot use the same to keep the experiments consistent and more comparable.

Response: We thank the reviewer for the most helpful review and comments. DT stands for diphtheria toxin, and this information was added in Line 175. We further utilized

TFPI^{ff};EpoR; *CD169^{DTR/+}* mice and treated *TFPI^{ff};EpoR* mice with clodronate liposomes to delete macrophages on *TFPI^{ff};EpoR* background in order to maintain consistency in our experiments. The data was added in Figure 2C-2G.

6. In figure 2F, it seems the macrophage depletion itself affected erythroid generation, the authors should analyze and explain it. Is this effect disappeared when TFPI is absent? The authors should do more work on this important issue.

Response: We agree with the reviewer's comment that macrophage depletion itself affected erythroid generation. Previous studies have reported the crucial involvement of macrophages in erythropoiesis under both steady-state and stress conditions. Depletion of macrophages, on one hand, leads to impaired physiological erythropoiesis and induced anemia^{11,42,43}. On the other hand, macrophage depletion prevents mice recovering from induced anemia and improves the phenotype of polycythemia vera⁴². This explanation and analysis were added in discussion, Line 405-409.

Clodronate liposome treatment resulted in a slight decrease in RBC numbers compared with control mice on *TFPI^{ff};EpoR* background ($P = 0.041$, $n = 10$), indicating that macrophage depletion still impaired erythroid generation when TFPI was absent. The result was added in Figure 2J and result description was added in Line 186-188.

We further expanded our investigation by utilizing a PHZ-induced stress erythropoiesis model to explore this issue, and discovered a delayed RBC recovery response of clodronate liposome-treated mice compared with control mice when TFPI was absent. The result was added in Figure 2L, and the result description was added in

Line 188-189.

Figure 2J. (J) PB RBC numbers, Hb, and HCT in $TFPI^{f/f;EpoR}$ mice after clodronate liposomes treatment.

Figure 2L. (L) PB RBC numbers, Hb, and HCT of $TFPI^{f/f;EpoR}$ mice after clodronate liposomes and PHZ treatment.

References:

11. Chow A, *et al.* CD169⁺ macrophages provide a niche promoting erythropoiesis under homeostasis and stress. *Nat Med* **19**, 429-436 (2013).
42. Ramos P, *et al.* Macrophages support pathological erythropoiesis in polycythemia vera and β -thalassemia. *Nat Med* **19**, 437-445 (2013).
43. Ramos P, *et al.* Enhanced erythropoiesis in Hfe-KO mice indicates a role for Hfe in the modulation of erythroid iron homeostasis. *Blood* **117**, 1379-1389 (2011).

7. In figure 2G, in the PBS control group, the percentage of erythroblast significantly increased in TFPI^{f/f} EpoR mice in stage III and IV, but dropped down in stage V, explain why.

Response: We thank the reviewer for this comment which helps us to improve our study. Macrophage depletion in mice with polycythemia vera results in a block in erythroblast differentiation at III and IV stages⁴², indicating the transition from III and IV stages to V stage is a key point in erythropoiesis. Heme production in erythroblasts reaches a maximum in the stage III and IV to promote hemoglobin production and erythroblast maturation^{51,52}. Here, we found that TFPI deficiency led to the inhibition of heme synthesis in central macrophages. Macrophages-specific knockout of Fech, the terminal heme synthesis enzyme, also prevented hemoglobin synthesis and blocked erythroblast differentiation at III and IV stages. Therefore, the inhibition of heme synthesis in central macrophages of *TFPI^{f/f}EpoR* mice results in a block at III and IV stages of erythroblast development. It demonstrates that central macrophage-derived heme also promotes hemoglobin production and erythroblast maturation. The discussion was added in Line 434-445, and the result description was added in Line 182-184.

References:

42. Ramos P, *et al.* Macrophages support pathological erythropoiesis in polycythemia vera and β -thalassemia. *Nat Med* **19**, 437-445 (2013).
51. Hoffman LM, Ross J. The role of heme in the maturation of erythroblasts: the effects of inhibition of pyridoxine metabolism. *Blood* **55**, 762-771 (1980).
52. Medlock AE, Dailey HA. New avenues of heme synthesis regulation. *Int J Mol Sci*

8. In Figure 3J, the authors claimed that ‘We found that the phosphorylation level of GATA1 in Ter119+ 207 cells and central macrophages increased over time’, but they only showed central macrophages and didn’t mention which mice model the macrophages were isolated from.

Response: We thank the reviewer for this comment which helps us to improve our study. We only investigated that TFPI increased the phosphorylation level of GATA1 in central macrophages. It was a mistake and ‘Ter119+ cells’ was deleted in the result, Line 242. The central macrophages were isolated from *TFPI^{fl/fl};EpoR* mice, and this information was added in the result, Line 243, and in Figure 3J.

9. In the results ‘TFPI interacts with Thbd in central macrophages’, the authors showed TFPI interacted with Thbd in BM cells by co-IP, but in order to confirm their hypothesis, central macrophages should be used instead.

Response: We agree with the reviewer’s comment. Co-IP experiment was performed in central macrophages and the result was added in Figure 4I.

Figure 4I. (I) Co-IP analysis of the interaction between TFPI and Thbd in

F4/80⁺CD169⁺Vcam-1⁺ macrophages.

Reviewer #2 (Remarks to the Author):

In this paper, Ma et al. found that TFPI, an anti-coagulant protein, regulates erythropoiesis via controlling heme biosynthesis in erythroblastic island (EBI) central macrophage. They used TFPI tissue-specific knock-out (KO) mice and polycythemia mouse models. The functions of TFPI were linked with EBI central macrophage. The authors found that TFPI on the erythroblast surface interacted with Thbd expressed on the central macrophage surface, and the interaction activated the p-ERK/GATA1/Fech pathway to control heme biosynthesis in the central macrophage. These findings were further proved with several mouse modes, including Thbd and Fech tissue-specific KO mice. Finally, they explore the applications of targeting TFPI to treat polycythemia. They concluded that TFPI from erythroblasts drives heme biosynthesis in central macrophage to support erythropoiesis in polycythemia and can be targeted for treating polycythemia.

While the manuscript presented numerous datasets using different model systems, the underlying logic is unclear. It is important to resolve the following concerns for the story to be published.

Response: We thank the reviewer's helpful and constructive suggestions for improvement. We have thoroughly examined each of the points raised and added relevant data and information to make the manuscript clearer.

The point-by-point responses to the reviewer's comments and questions are provided below.

1, Thrombosis in polycythemia is complex and involves platelets, red blood cells, and many other factors. TFPI is an inhibitor of coagulation. TFPI antibodies in clinical trials were evaluated based on TFPI's roles in inhibiting coagulation. From this perspective, inhibiting TFPI in polycythemia may further promote thrombosis and worsen the disease. In this study, what about the survival of those mice and their thrombosis treated with TFPI antibody in supplemental Figure 7? These data should be shown. The same questions apply to the JAK2^{V617F}-TFPI f/f VaV mice in Figure 1K-1L.

Response: We thank the reviewer for this comment which helps us to improve our study. TFPI mAb treatment increased thrombus size and weight but did not influence the survival rate of *Jak2*^{V617F}-mutated mice (supplemental Figure 7B and 7C), while erythroid lineage-specific TFPI deficiency increased thrombus size and weight but did not influence mice survival rate in *Jak2*^{V617F} mice (supplemental Figure 1F and 1G), consistent with previous study indicating that endothelial cells specific TFPI knockout only resulted in a relatively mild thrombosis phenotype³¹. The thrombus size and weight and survival curve information were added in supplemental Figure 7B and 7C, supplemental Figure 1F and 1G, and the description in Line 368-370 and 139-143.

supplemental Figure 7B and 7C. (B) Thrombus size and weight in *Jak2*^{V617F}-mutated mice and TFPI mAb-treated *Jak2*^{V617F}-mutated mice after IVC flow restriction. (C)

Kaplan-Meier survival analysis of *Jak2*^{V617F}-mutated mice (n = 11) and TFPI mAb-treated *Jak2*^{V617F}-mutated mice (n = 15).

supplemental Figure 1F and 1G. (F) Thrombus size and weight in *TFPI*^{ff} mice, *Jak2*^{V617F}-mutated mice, *TFPI*^{ff;Vav} mice, and *Jak2*^{V617F};*TFPI*^{ff;Vav} mice after IVC flow restriction. (G) Kaplan-Meier survival analysis of *TFPI*^{ff} mice (n = 10), *Jak2*^{V617F}-mutated mice (n = 13), *TFPI*^{ff;Vav} mice (n = 10), and *Jak2*^{V617F};*TFPI*^{ff;Vav} mice (n = 15).

Reference:

31. White TA, *et al.* Endothelial-derived tissue factor pathway inhibitor regulates arterial thrombosis but is not required for development or hemostasis. *Blood* **116**, 1787-1794 (2010).

2, Even though the data demonstrates that TFPI plays roles in erythropoiesis via TFPI-Thbd-mediated erythroblast-macrophage interaction, it does not exclude TFPI's intrinsic roles in erythropoiesis. TFPI's intrinsic roles in erythroblast should be included in the manuscript, and they can be monitored after knock-out or over-expression of TFPI in erythroblast without macrophage co-culture.

Response: We agree with the reviewer's comment. Ter119⁺CD71⁺ cells were cultured independently after TFPI knockout. The findings revealed that TFPI had no effect on

erythropoiesis in the absence of macrophages. The results were added in Figure 2A and 2B, and the result description was added in Line 164-169.

Figure 2A and 2B. (A) Experimental design to determine the effect of TFPI on the proliferation of Ter119⁺CD71⁺ cells *in vitro*. (B) The effect of erythroid lineage-specific TFPI knockout on the expansion of Ter119⁺CD71⁺ cells.

3, What about the EBI number changes in bone marrow and spleen of those mice, including TFPI KO mice, *Thbd* KO mice, and rTFPI treated mice? These results should be included to make their conclusions stronger.

Response: We thank the reviewer for this comment which helps us to improve our study. EBIs were isolated from bone marrow and spleen of *TFPI^{f/f}EpoR* mice, *Thbd^{f/f}LysM* mice, and rTFPI treated mice. The findings revealed a decrease in EBI numbers in *TFPI^{f/f}EpoR* and *Thbd^{f/f}LysM* mice, while an increase in rTFPI treated mice. The results were added in Figure 1L and 1M, Figure 5G and 5H, and supplemental Figure 2J and 2K, and the result description was added in Line136-137, 301-302, and 195-196.

Figure 1L and 1M. (L) EBI (F4/80⁺Ter119⁺ live multiplets) numbers in the BM of *TFPI*^{f/f;EpoR} mice. (M) EBI numbers in the spleen of *TFPI*^{f/f;EpoR} mice.

Figure 5G and 5H. (G) EBI numbers in the BM of *Thbd*^{f/f;LysM} mice. (H) EBI numbers in the spleen of *Thbd*^{f/f;LysM} mice.

supplemental Figure 2J and 2K. (J) EBI numbers in the BM of rTFPI-treated mice. (K) EBI numbers in the spleen of rTFPI-treated mice.

4, Many shRNAs were used in the study. ShRNAs commonly have off-target effects, CRISPER-CAS9 system is preferred. Even with the shRNAs, at least the western blot

showing the efficiency of those shRNAs should be included.

Response: We thank the reviewer for this comment which helps us to improve our study.

Western blot data was added in supplemental Figure 1C and supplemental Figure 2B for TFPI shRNA treatment, Figure 3K for GATA1 shRNA treatment, and Figure 4K for Thbd shRNA treatment. We further utilized TF monoclonal antibody to blockade TF, and added the data in Figure 4B-4E, data description in Line 269-274, and utilized intraosseous infusion of aPC shRNA to enhance the knockdown efficiency and added the data in Figure 6B and 6C, data description in Line 325-330.

supplemental Figure 1C. (C) TFPI protein expression of Ter119⁺ cells after TFPI shRNA treatment.

supplemental Figure 2B. (B) TFPI protein expression of Ter119⁺CD71⁺ cells after TFPI shRNA treatment.

Figure 3K. (K) GATA1 protein expression of F4/80⁺CD169⁺Vcam-1⁺ macrophages

after GATA1 shRNA treatment.

Figure 4K. (K) Thbd protein expression in F4/80⁺CD169⁺Vcam-1⁺ macrophages after Thbd shRNA treatment.

Figure 4B-4E. (B) TF procoagulant activity and plasma TAT levels in mice after TF mAb treatment (C) Heme content in F4/80⁺CD169⁺Vcam-1⁺ macrophages of *TFPI^{ff/f;EpoR}* mice after TF mAb treatment. (D) Fech mRNA expression in F4/80⁺CD169⁺Vcam-1⁺ macrophages of *TFPI^{ff/f;EpoR}* mice after TF mAb treatment. (E) PB Hb in *TFPI^{ff/f;EpoR}* mice after TF mAb treatment.

Figure 6B and 6C. (B) aPC protein expression of F4/80⁺CD169⁺Vcam-1⁺ macrophages after intraosseous infusion of lentivirus-expressing aPC shRNA. (C) Frequency of RIII, RIV, and RV erythroblast populations in *Thbd^{ff/f;LysM}* mice after intraosseous infusion of lentivirus-expressing aPC shRNA.

5, in Figure 1A, TFPI expression levels were not higher in JAK2V617F positive samples compared to normal cases, which needs to be explained.

Response: We agree with the reviewer's comment. TFPI expression levels were not higher in plasma of *JAK2^{V617F}* positive samples compared to normal cases, consistent with previous work³⁹. However, a higher TFPI expression level was found in *JAK2^{V617F}*-mutated erythroblasts. Therefore, erythroblast-derived TFPI may play a local role in erythropoiesis in bone marrow. The discussion in Line 390-394.

Reference:

39. Marchetti M, *et al.* Thrombin generation and activated protein C resistance in

patients with essential thrombocythemia and polycythemia vera. *Blood* **112**, 4061-4068 (2008).

6, in Figure 1D, the difference in TFPI expression levels seems related to JAK2V617F mutation, other than co-culture of macrophage and erythroblast.

Response: We agree with reviewer's comment that the difference in TFPI expression levels of erythroblasts related to JAK2V617F mutation, other than co-culture with macrophages. We further cultured erythroblasts independently without macrophages and revealed that co-culturing erythroblasts with macrophages did not influence the expression levels of TFPI in erythroblasts. The result was added in Figure 1E, and the result description was added in Line 113-114.

Figure 1E. (E) TFPI mRNA expression in erythroblasts cultured with or without macrophages. Alone, erythroblasts culture independently; co-culture, erythroblasts co-culture with macrophages.

7, in Figure 1I, apoptotic cell percentages in the control group are more than 60%. The data is not reliable and should be repeated. The same problems are in supplemental Figure 3E, supplemental Figure 4I, and Figure 5F.

Response: We thank the reviewer for this comment which helps us to improve our study.

We systematically adjusted the cell quantity and antibody dosage, and repeated the experiment. The result was changed in Figure 1J, supplemental Figure 3E, supplemental Figure 4I and Figure 5F.

Figure 1J. (J) Frequency of apoptotic erythroblasts among BM cells in *TFPI*^{f/f;EpoR} mice.

supplemental Figure 3E. (E) Frequency of apoptotic erythroblasts among BM cells in *TFPI*^{f/f;CD169} mice.

supplemental Figure 4I. (I) Frequency of apoptotic erythroblasts among BM cells in *Fech*^{f/f;CD169} mice.

Figure 5F. (F) Frequency of apoptotic erythroblasts among BM cells in *Thbd*^{f/f;LysM} mice.

8, in Figure 2A-2B, TFPI KO cells, instead of shRNA-mediated knock-down, are better to be used since TFPI tissue-specific KO mice are available, and the KO cells can be purified directly from mice.

Response: We thank the reviewer for the most helpful review and comments. TFPI KO cells were purified from *TFPI*^{f/f;EpoR} mice and erythroblast proliferation assay was then performed. The data was added in Figure 2A and 2B, the data description was added in Line 164-169.

Figure 2A and 2B. (A) Experimental design to determine the effect of TFPI on the proliferation of Ter119⁺CD71⁺ cells *in vitro*. (B) The effect of erythroid lineage-specific TFPI knockout on the expansion of Ter119⁺CD71⁺ cells.

9, In Figure 2D, the picture on the right is not consistent with the three left statistic figures since it looks like there is a significant difference between the third and fourth tubes.

Response: We agree with the reviewer's comment. We repeated this experiment and added the results in Figure 2G.

Figure 2G. (G) BM cell pellets from *TFPI^{f/f}EpoR*; *CD169^{DTR/+}* mice after DT treatment.

10, Was TF expressed on central macrophage? In Figure 4A, TF expression levels should be tested on central macrophage instead of total bone marrow. Similarly, in Figure 6A, aPC mRNA levels should be tested in different lineage cells instead of total bone marrow since total bone marrow is a mixture of cells.

Response: We agree with the reviewer's comment. TF was expressed in central macrophages. TF and aPC expression levels were tested on central macrophages instead, and the data was added in Figure 4A and Figure 6A.

Figure 4A (A) mRNA expression of TF in F4/80⁺CD169⁺Vcam-1⁺ macrophages of *Jak2*^{V617F}-mutated or hypoxia-exposed mice.

Figure 6A. (A) mRNA expression of aPC in F4/80⁺CD169⁺Vcam-1⁺ macrophages of *Jak2*^{V617F}-mutated and hypoxia-exposed mice.

11, in Figure 6E, the significance of comparing the second and fourth groups should be shown in the figure to support their conclusion “SCH772984 treatment resulted in no further reduction in heme content after GATA1 shRNA treatment (Figure 6E)”.

Response: We agree with the reviewer's comment. We added experimental samples and illustrated the significance between the second and fourth groups in Figure 6F. The

second and fourth groups showed no significance ($P = 0.166$, $n = 10$).

Figure 6F. (F) Heme content in $F4/80^+CD169^+Vcam-1^+$ macrophages after treated with rTFPI combined with ERK1/2 inhibitor and GATA1 shRNA.

12, in Figure 6G and 6H, the blot of p-GATA1 should be repeated since the current data do not support their conclusion “We then found that the protein level of p-GATA1 was decreased in *Thbd* *f/f*; *LysM* or SCH772984-treated mice”.

Response: We agree with the reviewer’s comment. The blot was repeated and the data was put in Figure 6H and 6I.

Figure 6H and 6I. (H) Phosphorylation level of GATA1 protein in $F4/80^+CD169^+Vcam-1^+$ macrophages of *Thbd*^{*f/f*; *LysM*} mice after rTFPI treatment. (I) Phosphorylation level of GATA1 protein in $F4/80^+CD169^+Vcam-1^+$ macrophages treated with rTFPI and ERK1/2 inhibitor.

13, in Figure 6L, it seems that the fourth group is higher than the second group, which

do not support their conclusion “this function was abrogated by SCH772984 and GATA1 shRNA treatment. (Figure 6J-6L)”. The significance of comparing the second and fourth groups should be shown in the figure.

Response: We thank the reviewer for this comment which helps us to improve our study. In the data from the firstly submitted manuscript, the rTFPI treatment (the fourth group) exhibited higher ALAS2 and Fech levels than control (the second group) after GATA1 shRNA treatment (left $P = 0.016$, right $P = 0.008$). The significance was shown in Figure 6M, and the sentence “this function was abrogated by SCH772984 and GATA1 shRNA treatment” was corrected to “this effect was attenuated by SCH772984 and GATA1 shRNA treatment” in Line 350-351.

14, The working model in Figure 6M showed that GATA1 binds the gene and promotes Fech’s expression. It seems that GATA1 is a direct transcription factor of Fech. The original study should be cited if this finding was published before, or data should be shown to prove it if it was found by the authors.

Response: We thank the reviewer for this comment which helps us to improve our study. We cited the original study that GATA1 is a direct transcription factor of Fech³⁶, and added it in Line 241, reference 36.

Reference:

36. Taketani S, Mohri T, Hioki K, Tokunaga R, Kohno H. Structure and transcriptional regulation of the mouse ferrochelatase gene. *Gene* **227**, 117-124 (1999).

15. Currently, the identification of EBI macrophage is not fully understood. The marker combination Ly6G/Ter119⁺F4/80⁺CD169⁺Vcam-1⁺ used in this manuscript cannot represent all EBI macrophages, so the statement of “central macrophage” to refer to the sorted macrophages is not accurate.

Response: We agree with the reviewer’s comment that Ly6G/Ter119⁺F4/80⁺CD169⁺Vcam-1⁺ cannot represent all EBI macrophages. Ly6G/Ter119⁺F4/80⁺CD169⁺Vcam-1⁺ cells were sorted to represent central macrophage-enriched cells in our study^{12,22,30}. The paragraph was added in Line 119-121.

We further systematically changed “central macrophage” to “F4/80⁺CD169⁺Vcam-1⁺ macrophage” in Line 122, 174, 201 and elsewhere when referring to our sorted macrophages to make our study more accurate.

References:

12. Li W, *et al.* Identification and transcriptome analysis of erythroblastic island macrophages. *Blood* **134**, 480-491 (2019).
22. Yang C, *et al.* Mitochondria transfer mediates stress erythropoiesis by altering the bioenergetic profiles of early erythroblasts through CD47. *J Exp Med* **219**, e20220685 (2022).
30. Wculek SK, Dunphy G, Heras-Murillo I, Mastrangelo A, Sancho D. Metabolism of tissue macrophages in homeostasis and pathology. *Cell Mol Immunol* **19**, 384-408 (2022).

REVIEWERS' COMMENTS

Reviewer #1 (Remarks to the Author):

The authors have addressed all my concerns.

Reviewer #2 (Remarks to the Author):

The authors have adequately addressed all my comments.